# UnIVAL: Unified Model for Image, Video, Audio and Language Tasks

**Mustafa Shukor**                                    *mustafa.shukor@sorbonne-universite.fr*
*Sorbonne University*
*MLIA, ISIR, Paris, France*

**Corentin Dancette**                                *corentin.dancette@sorbonne-universite.fr*
*Sorbonne University*
*MLIA, ISIR, Paris, France*

**Alexandre Rame**                                    *alexandre.rame@isir.upmc.fr*
*Sorbonne University*
*MLIA, ISIR, Paris, France*

**Matthieu Cord**                                    *matthieu.cord@sorbonne-universite.fr*
*Sorbonne University, MLIA, ISIR, Paris, France*
*Valeo.ai, Paris, France*

**Reviewed on OpenReview:** *https://openreview.net/forum?id=4uflhObpcp*

## Abstract

Large Language Models (LLMs) have made the ambitious quest for generalist agents significantly far from being a fantasy. A key hurdle for building such general models is the diversity and heterogeneity of tasks and modalities. A promising solution is unification, allowing the support of a myriad of tasks and modalities within one unified framework. While few large models (*e.g.*, Flamingo (Alayrac et al., 2022)), trained on massive datasets, can support more than two modalities, current small to mid-scale unified models are still limited to 2 modalities, usually image-text or video-text. The question that we ask is: *is it possible to build efficiently a unified model that can support all modalities?* To answer this, we propose **UnIVAL**, a step further towards this ambitious goal. Without relying on fancy datasets sizes or models with billions of parameters, the $\sim 0.25$B parameter **UnIVAL** model goes beyond two modalities and unifies text, images, video, and audio into a single model. Our model is efficiently pretrained on many tasks, based on task balancing and multimodal curriculum learning. **UnIVAL** shows competitive performance to existing state-of-the-art approaches, across image and video-text tasks. The feature representations learned from image and video-text modalities, allows the model to achieve competitive performance when finetuned on audio-text tasks, despite not being pretrained on audio. Thanks to the unified model, we propose a novel study on multimodal model merging via weight interpolation of models trained on different multimodal tasks, showing their benefits in particular for out-of-distribution generalization. Finally, we motivate unification by showing the synergy between tasks. The model weights and code are available at: https://github.com/mshukor/UnIVAL.

## 1 Introduction

The advent of Large Language Models (LLMs) (Brown et al., 2020; Rae et al., 2021; Chowdhery et al., 2022; Tay et al., 2022) represents a significant step towards the development of generalist models. Generally based on the Transformer architecture (Vaswani et al., 2017) and a single next-token prediction objective, they continue to astound the world with their remarkable performances in text understanding and generation.

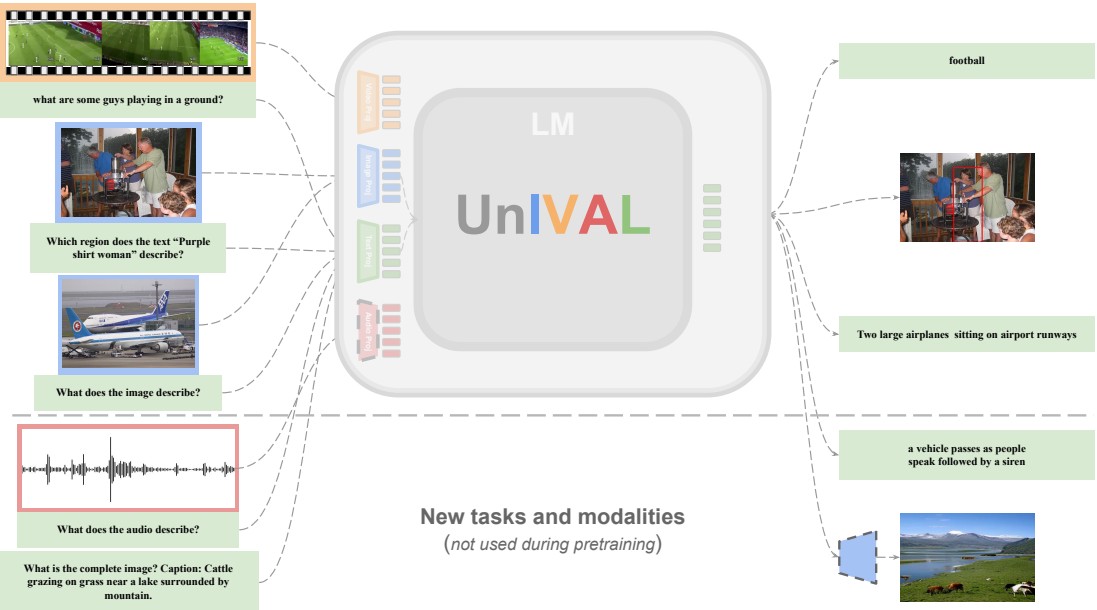

Figure 1: **Un**IVAL model. Our sequence-to-sequence model unifies the architecture, tasks, input/output format, and training objective (next token prediction). **Un**IVAL is pretrained on image and video-text tasks and can be finetuned to tackle new modalities (audio-text) and tasks (text-to-image generation) that were not used during pretraining.

Nevertheless, their current limitation to a single modality (text) restricts their understanding and interaction with the world. This highlights the need for robust multimodal models handling diverse tasks across numerous modalities. Recently, many works have tried to go beyond single modality, and build powerful multimodal models (Huang et al., 2023; Driess et al., 2023; Li et al., 2023) that surpass previous task/modality-specific approaches. However, most of these works focus on image-text tasks and only a handful of approaches aim to incorporate more than two modalities, such as image/video-text (Alayrac et al., 2022; Wang et al., 2022b).

The prevailing approach for pretraining multimodal models revolves around training them on large, noisy image-caption datasets (Schuhmann et al., 2021; Jia et al., 2021; Radford et al., 2021), where the model is tasked with generating or aligning image-captions through causal generation or unmasking. However, this approach encounters a significant challenge: it relies on extensive datasets to compensate for the inherent noise and the relatively simple task of caption generation. In contrast, multitask learning (Caruana, 1997) on relatively small yet high-quality datasets presents an alternative solution to learn efficient models capable of competing with their large-scale counterparts (Alayrac et al., 2022; Chen et al., 2022b; Reed et al., 2022).

Current small to mid-scale (less than couple of hundred million parameters) vision-language models (Li et al., 2019; Shukor et al., 2022; Dou et al., 2021; Li et al., 2022b) still have task-specific modules/heads, many training objectives, and support a very small number of downstream tasks due to the different input/output format. Recently, the sequence-to-sequence OFA (Wang et al., 2022c) and Unified-IO (Lu et al., 2022a) have made a noticeable step towards more unified systems that can support a wide range of image and image-text tasks, with more reasonable scales (*e.g.* can fit on user-grade GPU). These models are pretrained on many good quality, public benchmarks. On video-text tasks, LAVENDER (Li et al., 2022c) takes a similar direction by unifying the pretraining tasks as Masked Language Modeling (MLM). Sequence-to-sequence unified models are particularly well-suited for open-ended text generation tasks and can readily incorporate recent LLMs. To guide the model in solving a specific task, a textual prompt resembling an instruction (Raffel et al., 2020) is added at the beginning of the input sequence. They have the capability to unify tasks across different modalities, and thus easily supporting new tasks, by representing all inputs and outputs as sequences of tokens, utilizing an unified input/output format and vocabulary. These tokens can represent various modalities such as text, image patches, bounding boxes, audio, video, or any other modality, without

the need for task-specific modules/heads. These strategies are straightforward to scale and manage, as they involve a single training objective and a single model.

However, existing works are still limited to downstream tasks with no more than 2 modalities (image-text or video-text). Providing unification across a larger number of tasks and modalities would offers additional advantages. First, we would benefit from the knowledge transfer across them, by harnessing their collaborative strengths. Second, once pretraining is done, the model can be finetuned on many different datasets: because of the wider range of more diverse pretraining data, unification across more tasks would enable better and more efficient generalization after finetuning on novel tasks and modalities. In this paper, we thus ask the following question.

*Is it possible to efficiently build a unified model that can support all modalities?*

A positive answer to this question will pave the way for building generalist models that can potentially solve any task. To answer this question, we propose **UnIVAL**, a step further towards generalist modality-agnostic models. **UnIVAL** (illustrated in Fig.1) goes beyond two modalities and unifies text, images, video, and audio into a single model.

Our contributions are multiple:

- To the best of our knowledge, **UnIVAL** is the first model, with unified architecture, vocabulary, input/output format, and training objective, that is able to tackle image, video, and audio language tasks, without relying on large scale training or large model size. Our 0.25B parameter model achieves competitive performance to existing modality-customized work. With comparable model sizes, we achieves new SoTA on some tasks (*e.g.* +1.4/+0.98/+0.46 points accuracy on RefCOCO/RefCOCO+/RefCOCOg Visual Grounding, +3.4 CIDEr on Audiocaps) .

- We show the benefits of multimodal curriculum learning with task balancing, for efficiently training the model beyond two modalities.

- We show the importance of multitask pretraining, compared to the standard single task one, and study the knowledge transfer between pretrained tasks and modalities. In addition, we find that pretraining on more modalities makes the model generalizes better to new ones. In particular, without any audio pretraining, **UnIVAL** is able to attain competitive performance to SoTA when finetuned on audio-text tasks.

- We propose a novel study on multimodal model merging via weight interpolation (Izmailov et al., 2018; Wortsman et al., 2022; Rame et al., 2022). We show that, when weights are finetuned on different multimodal tasks from our unified pretrained model, interpolation in the weight space can effectively combine the skills of the various finetuned weights, creating more robust multitask models without any inference overhead. Thus, in addition to multitask pretraining, averaging differently finetuned weights is another way to leverage and recycle (Rame et al., 2023a) the diversity of multimodal tasks, enabling their collaboration. This is the first study of weight interpolation showing its effectiveness with multimodal foundation models.

## 2 Related Work

We provide a brief related work, further detailed in Appendix B.

**Multimodal pretraining.** So far, most of the efforts to build multimodal models have been focused on vision-language pretraining. Contrastive-based approaches (Radford et al., 2021; Jia et al., 2021) try to learn a shared and aligned latent space by training on hundreds of millions of pairs. More data-efficient approaches (Shukor et al., 2022; Li et al., 2021a; 2022b; Dou et al., 2021; Singh et al., 2022) relied on additional multimodal interaction modules and variety of training objectives such as image-text matching, masked language modeling and image-text contrastive (Chen et al., 2020c; Kim et al., 2021; Lu et al., 2019; Zhang et al., 2021). In the video-language community, similar approaches have tried to model the interaction

between language and frames sequences (Cheng et al., 2022; Wang et al., 2023a; Fu et al., 2021; Zellers et al., 2021; Yang et al., 2021a). Few works have targeted both image and video language pretraining (Wang et al., 2022b).

| Method | PT examples. I (V) | Model Size | Param. init | | PT Modalities | | DS Modalities | | | Unified | | | |
|---|---|---|---|---|---|---|---|---|---|---|---|---|---|
| | | | V | L | I-T | V-T | I-T | V-T | A-T | Arch. | I/O | Tasks | Objective |
| GIT/2 (Wang et al., 2022a) | 0.8B/12.9B | 0.7B/5.1B | Florence/DaViT | Random | ✓ | | ✓ | ✓ | | | | ✓ | ✓ |
| PaLI (Chen et al., 2022b) | 12B+ | 3B/15B/17B | ViT-G | mT5 | ✓ | | ✓ | | | | | ✓ | ✓ |
| CoCa (Yu et al., 2022) | 4.8B | 2.1B | Random | Random | ✓ | | ✓ | ✓ | | | | | ✓ |
| Unified-IO (Lu et al., 2022a) | 130M+ | 0.2B/0.8B/2.8B | Random | T5 | ✓ | | ✓ | | | ✓ | ✓ | ✓ | ✓ |
| OmniVL (Wang et al., 2022b) | 15.3M (2.8M) | 0.2B | TimeSformer | BERT | ✓ | ✓ | ✓ | ✓ | | | | ✓ | |
| VIOLET (Fu et al., 2021) | 3.3M (182.5M) | 0.2B | VideoSwin | BERT | ✓ | ✓ | | ✓ | | ✓ | | | |
| Merlot Reserve (Zellers et al., 2022) | (960M) | ∼ 0.3B/0.7B | ViT/AST | - | | ✓ | | ✓ | | | | ✓ | ✓ |
| LAVENDER (Li et al., 2022c) | 19M (14.4M) | ∼ 0.2B | VidSwin | BERT | ✓ | ✓ | | ✓ | | | | ✓ | ✓ |
| BLIP-2 (Li et al., 2023) | 129M+ | 12.1B | EVA/CLIP | FlanT5/OPT | ✓ | | ✓ | | | | | | ✓ |
| FLamingo (Alayrac et al., 2022) | 2.3B (27M) | 3.2B/9.3B/80B | CLIP | Chinchilla | ✓ | ✓ | ✓ | ✓ | | | | ✓ | ✓ |
| OFA (Wang et al., 2022c) | 60M+ | 0.2B/0.5B/0.9B | ResNet | BART | ✓ | | ✓ | | | ✓ | ✓ | ✓ | ✓ |
| Gato (Reed et al., 2022) | 2.2B+ | 1.2B | ResNet | N/A | ✓ | | ✓ | | | ✓ | ✓ | ✓ | ✓ |
| UnIVAL (ours) | 21.4M (5M) | 0.25B | ResNet/ResNeXt | BART | ✓ | ✓ | ✓ | ✓ | ✓ | ✓ | ✓ | ✓ | ✓ |

Table 1: Comparison of different foundation models. Our **UnIVAL** approach is pretrained on a relatively small dataset, tackles image/video/audio-text modalities, while unifying the 4 different aspects explained in Sec.3 and in Appendix C: unified model, input/output format, pretraining tasks, and training objective.

**Unified models.** Building unified systems has been explored first in the NLP community. Raffel et al. (2020) proposed the T5 transformer model, a text-to-text framework that solves many NLP tasks, each one being described by a task-specific textual prefix. Since then, building general textual models has been heavily explored with LLMs (Brown et al., 2020; Rae et al., 2021; Chowdhery et al., 2022). This inspired other communities to build unified models. In the vision community, the work of Chen et al. (2022a), proposed a pixel-to-sequence framework to unify different vision tasks such as object detection and instance segmentation. For multimodal tasks, Cho et al. (2021) proposed to unify vision-language tasks as conditional text generation. OFA (Wang et al., 2022c) then proposed a large-scale sequence-to-sequence framework and extended previous approaches to more image-text tasks, including text-to-image generation. Similarly, Unified-IO (Lu et al., 2022a), in addition to image-text tasks, targets many visual tasks including dense prediction ones. The closest works to us are indeed OFA (Wang et al., 2022c) and Unified-IO (Lu et al., 2022a), however, we propose to unify tasks across more modalities, with significantly smaller model and dataset sizes. The differences are clarified in Tab.1, where we compare different foundation models involving unification.

**Weight averaging across multimodal tasks.** To combine multiple expert models with diverse specializations, we leverage a simple yet practical strategy: *linear interpolation in the weight space.* We follow Ilharco et al. (2023); Daheim et al. (2023); Ortiz-Jimenez et al. (2023) suggesting that averaging networks in weights can combine their abilities without any computational overhead. In particular, weight averaging (WA) was shown useful in model soups approaches (Wortsman et al., 2022; Rame et al., 2022) to improve out-of-distribution generalization as an approximation of the more costly averaging of predictions (Lakshminarayanan et al., 2017). Recent works extended WA to weights fine-tuned with different losses (Rame et al., 2022; 2023b; Croce et al., 2023) or on different datasets (Matena & Raffel, 2022; Choshen et al., 2022; Don-Yehiya et al., 2023; Rame et al., 2023a). In addition, some techniques try to leverage the features learned on different auxiliary tasks for a given target task. Fusing (Choshen et al., 2022) averages multiple auxiliary weights to serve as an initialization for the unique finetuning on the target task. In contrast, ratatouille (Rame et al., 2023a) delays the averaging after the multiple finetunings on the target tasks: each auxiliary model is finetuned independantly on the target task, and then all fine-tuned weights are averaged. These approaches consider classification tasks, for a given modality (usually images): interpolating weights of models trained on different multimodal tasks is very little investigated. The most similar and concurrent work is the recent Sung et al. (2023) applying a complex architecture-specific merging strategy. This work differs from us, as we explore WA during finetuning on multimodal downstream tasks, where they merge models pretrained on different modalities.

# 3 Pretraining of UnIVAL

Current multimodal models are pretrained on massive noisy datasets with a limited number of tasks (*e.g.*, image-conditioned text generation). We focus on the challenge of achieving reasonable performance without relying on vast amounts of data. Our approach involves multi-task pretraining on many good-quality datasets. We hope that the quality mitigates the need for massive datasets, thereby reducing computational requirements, while enhancing the overall model capability. The adoption of such an approach has become increasingly easy due to the growing availability of public, human-annotated, or automatically generated datasets. UnIVAL is unified along the following 4 axes (more detailed in Appendix C); model, input/output format, pretraining tasks, and training objective.

## 3.1 Unified model

Our model's core is a LM designed to process abstract representations. It is enhanced with lightweight modality-specific projections that enable the mapping of different modalities to a shared and more abstract representation space, which can then be processed by the LM. We use the same model during pretraining and finetuning of all tasks, without any task-specific heads. We detail below key components of this architecture, that are further detailed in Appendix D.

**Shared module.** To tackle multimodal tasks at small to mid-scale, we employ an encoder-decoder LM, as its effectiveness for multimodal tasks has been demonstrated compared to decoder-only LMs (Wang et al., 2021), in addition to its superiority in zero-shot generalization after multitask training (Wang et al., 2022d). Another advantage of this architecture is the inclusion of bidirectional attention mechanisms in addition to unidirectional causal attention. This is particularly beneficial for processing various non-textual modalities. Our model accepts a sequence of tokens representing different modalities as input and generates a sequence of tokens as output.

**Light-weight specialized modules.** To optimize data and compute requirements, it is crucial to map different modalities to a shared representation space, before feeding them into the encoder of the LM. To achieve this, we employ lightweight modality-specific encoders. Each encoder extracts a feature map, which is then flattened to generate a sequence of tokens. These tokens are linearly projected to match the input dimension of the LM. It is important to strike a balance in the choice of encoder complexity. Using overly simplistic encoders, such as linear projections, may disrupt the LM, impede training speed, and necessitate larger datasets and then computational resources. Conversely, employing excessively complex encoders can hinder the benefits of learning a unified representation in the shared module. In our approach, we opt for CNN encoders as they scale effectively with high-resolution inputs, minimize the number of output tokens, and exhibit improved efficiency during both inference and training compared to transformers.

## 3.2 Unified input/output format

The input/output of all tasks consists of a sequence of tokens, where we use a unified vocabulary that contains text, location, and discrete image tokens.

## 3.3 Unified pretraining tasks

To train a single model on many tasks, a unified representation of these tasks is necessary. As our model's core is a LM, we transform all tasks into a sequence-to-sequence format, where each task is specified by a textual prompt (*e.g.*, "What does the video describe?" for video captioning).

For pretraining tasks, we pretrain only on relatively small public datasets, such as image captioning (COCO (Lin et al., 2014), Visual Genome (VG) (Krishna et al., 2017b), SBU (Ordonez et al., 2011), CC3M (Sharma et al., 2018) and CC12M (Changpinyo et al., 2021) (only in the first stage)), VQA (VQAv2 (Goyal et al., 2017), GQA (Hudson & Manning, 2019), VG (Krishna et al., 2017b)), Visual Grounding (VGround) and referring expression comprehension (RefCOCO, RefCOCO+, RefCOCOg (Yu et al., 2016)), video captioning

(WebVid2M (Bain et al., 2021)) and video question answering (WebVidQA (Yang et al., 2021a)). Note that we only use the training sets during pretraining. Pretraining tasks are further detailed in Appendix E.

### 3.4 Unified training objective

We follow other approaches (Wang et al., 2022c; Alayrac et al., 2022) and optimize the model for conditional next token prediction. Specifically, we use a cross-entropy loss.

### 3.5 Efficient pretraining

Besides unification across tasks and modalities, we detail different techniques for efficient pretraining.

**Multimodal curriculum learning (MCL).** Other works train the model on all tasks and modalities simultaneously (Wang et al., 2022c; Li et al., 2022c). However, we have observed that models trained on more modalities tend to exhibit better generalization to new ones. To capitalize on this, we employ a different strategy wherein we gradually introduce additional modalities during training. This approach facilitates a smoother transition to new modalities by providing a better initialization. Furthermore, this paradigm significantly reduces computational requirements compared to training on the entire dataset at once. Previous studies (Wang et al., 2022b) have demonstrated notable performance enhancements when employing this paradigm for shared visual encoders (applied to both images and videos). In our work, we extend this setting beyond shared visual encoders, and show its effectiveness for modality-specific projections and unified models. This approach mainly yields gains in training efficiency. This is important as it allows us to leverage existing pretrained multimodal models to incorporate new modalities. To validate the approach, we train the same model on image-text and video-text data for 20 epochs using 2 training approaches; the one-stage approach where we train on all data from the beginning, and our 2-stage curriculum training where we start to train on image-text for 10 epochs then we continue training on all data for the next 10 epochs. Tab.2, shows that the performance of both approaches are comparable. However, the 2-stage approach is more efficient in terms of training time (18% faster) and memory (25% less GPU memory).

| Method | Training Time | batch size (avg) | COCO (CIDEr) | VQAv2 (Acc) | RefCOCO+ (Acc@0.5) | MSR-VTT (CIDEr) | MSRVTT-QA (Acc) |
|--------|---------------|------------------|--------------|-------------|--------------------|-----------------|-----------------|
| One-stage | 2h04m | 4K | 127.9 | 73.21 | 70.89 | 55.9 | 42.38 |
| MCL | 1h42m | 3K | 128 | 73.24 | 70.19 | 56.3 | 42.27 |

Table 2: **Multimodal Curriculum learning (MCL).** We show that our multi-stage training is more efficient than the one stage one and leads to on par results. The training time is for one epoch on the same number of GPUs.

**Multimodal task balancing.** Contrary to previous work (Wang et al., 2022c), we find it more beneficial to balance the tasks in the batch, especially when using highly unbalanced datasets. Tab.3 shows some results. We compare models trained without balancing, where in each batch the number of examples for each task is proportional to the corresponding dataset size, and with task balancing, where the tasks have similar number of examples. The results show a consistent improvement after balancing especially with highly unbalanced datasets (*e.g.*, when adding CC12M, the overall performance drops significantly (B+CC12M)).

**Implementation details for pretraining.** The architecture of the LM is a typical encoder-decoder transformer initialized by BART-base (Lewis et al., 2020) with few modifications, following the implementation details of other work (Wang et al., 2022c). The modality-specific encoders are ResNet-101 pretrained on ImageNet as image encoder, 3D ResNext-101 (Hara et al., 2018b) pretrained on kinetics 400 as video encoder and PANN encoder pretrained for audio classification as audio encoder, we do not skip the last block as done by previous approaches (Wang et al., 2022c). We use Adam optimizer with weight decay 0.01 and linear decay scheduler for the learning rate starting from $2e-4$. All model parameters are pretrained in 2 stages; first we train only on image-text tasks for 150k steps and batch size 3200, then we add video-text tasks and continue training (after removing CC12M) for 90K steps with batch size 4K (2k for each modality). At the end of the

| Data | Task Balancing | COCO (CIDEr) | VQA v2 (Acc) | RefCOCO+ (Acc@0.5) |
|------|:---:|:---:|:---:|:---:|
| B | ✗ | 127.0 | 72.93 | 66.03 |
| B+CC12M | ✗ | 126.8 | 72.79 | 68.04 |
| B+VQA+Ground. | ✗ | 129.9 | 74.43 | 78.78 |
| B+VQA+Ground. | ✓ | 130.3 | 75.44 | 78.99 |
| B+VQA+Ground.+CC12M | ✗ | 129.9 | 75.21 | 78.85 |
| B+VQA+Ground.+CC12M | ✓ | 131.3 | 75.34 | 79.47 |

Table 3: **Multimodal task balancing.** Task balancing significantly improve the performance, especially when using datasets that largely differ in size (*e.g.*, CC12M). The baseline (B) consists of; VQAv2, RefCOCO+/CC3M/SBU/COCO/VG. VQA; GQA/VG. Ground.: RefCOCO/RefCOCOg.

last stage, we train the model for additional epoch after increasing the resolution of images from 384 to 480 and the videos from $224 \times 224$ and 8 frames to $384 \times 384$ and 16 frames. More details in Appendix G.

| Data (Modality) | Data size (# of examples) | Method | COCO (CIDEr) | VQAv2 (Acc) | RefCOCO+ (Acc@0.5) | MSR-VTT (CIDEr) | MSRVTT-QA (Acc) |
|---|:---:|:---:|:---:|:---:|:---:|:---:|:---:|
| CC3M (I) | 2.8M | | 117.3 | 69.5 | 55.2 | - | - |
| CC12M (I) | 10M | One-task pretraining | 120.2 | 71.6 | 56.7 | - | - |
| CC3M+CC12M (I) | 12.8M | | 123.6 | 71.7 | 59.8 | - | - |
| COCO+SBU+VG+CC3M (I) | 5M | | 125.8 | 72.0 | 56.1 | - | - |
| B (I) | 5.6M | | 127.0 | 72.9 | 66.0 | - | - |
| B+VQA (I) | 7.94M | | 128.9 | 73.2 | 71.0 | - | - |
| B+Ground (I) | 9.3M | Multitask pretraining | 129.8 | 74.4 | 77.6 | - | - |
| B+VQA+Ground (I) | 11.6M | | 129.9 | 75.1 | 78.8 | - | - |
| B+VQA+Ground+CC12M (I) | 21.6M | | 130.0 | 75.2 | 78.9 | - | - |
| B (I+V) | 8.1M | | 128.8 | 73.2 | 70.1 | 54.6 | 42.1 |
| B+WebVidQA (I+V) | 10.6M | Multitask pretraining | 128.0 | 73.2 | 70.2 | 56.3 | 42.3 |
| B+VQA+WebVidQA (I+V) | 13.9M | | 131.7 | 75.0 | 77.9 | 57.0 | 42.6 |
| B+Ground.+WebVidQA (I+V) | 17.6M | | 131.1 | 75.1 | 78.1 | 56.2 | 42.5 |

Table 4: **Knowledge transfer across tasks and datasets.** We show the synergy between different tasks and datasets. Multitask learning is efficient as it leverages the collaboration across tasks. Models are trained longer on I+V tasks.

**Knowledge transfer across tasks and modalities.** We investigate the knowledge transfer between tasks/modalities. We train for 10 epochs on image-text (I) datasets, followed by 10 epochs on image/video-text (I+V) datasets. The results are shown in Tab.4. We first compare between single and multitask learning. For single task, the models are trained on different image captioning datasets. For multitask learning, the models are trained for several tasks such as captioning, VQA or grounding. Overall, multitask learning is more efficient. as with comparable number of examples, it significantly outperforms models trained on single task. Second, we investigate the synergy between tasks and datasets. For image-text pretraining, there is a clear benefit of multitask training. Specifically, training on VQA helps to get +1.9 points on Captioning and +5 points for Visual Grounding. Similarly training on VGround, we have larger improvements on captioning and VQA. For image-text and video-text pretraining, VideoQA helps Video Caption and interestingly, Image VQA helps video tasks. We noticed that large datasets like CC12M does not bring significant improvements, compared to adding additional task with smaller number of examples. This also demonstrates that multitask learning is more efficient than large-scale single task learning.

We put in Appendix I our experiments that study further the **knowledge transfer across modalities.**

## 4 UnIVAL on Downstream Tasks

In this section, we present the experimental results of **UnIVAL** following different setups; finetuning on downstream datasets and direct evaluation without finetuning (*e.g.* zero-shot). As **UnIVAL** is unified and targets more than two modalities, for fair comparison to, we highlight other unified approaches in yellow , and models targeting more than 2 modalities in red . We did not highlight **UnIVAL** for clarity of presentation.

### 4.1 Finetuning on multimodal tasks

For **downstream tasks**, we finetune on standard image-text, video-text and audio-text benchmarks (Appendix G contains more implementation details). To have a fairer comparison with OFA, we finetune the author's released checkpoint (denoted as $OFA^{\dagger}_{Base}$) using the same hyperparametres as **UnIVAL**.

#### 4.1.1 Image-text tasks

| Model | RefCOCO | | | RefCOCO+ | | | RefCOCOg | |
|---|---|---|---|---|---|---|---|---|
| | val | testA | testB | val | testA | testB | val-u | test-u |
| VL-T5 (Cho et al., 2021) | - | - | - | - | - | - | - | 71.3 |
| UNITER (Chen et al., 2020c) | 81.41 | 87.04 | 74.17 | 75.90 | 81.45 | 66.70 | 74.86 | 75.77 |
| VILLA (Gan et al., 2020) | 82.39 | 87.48 | 74.84 | 76.17 | 81.54 | 66.84 | 76.18 | 76.71 |
| MDETR (Kamath et al., 2021) | 86.75 | 89.58 | 81.41 | 79.52 | 84.09 | 70.62 | 81.64 | 80.89 |
| UniTAB (Yang et al., 2021b) | 88.59 | 91.06 | 83.75 | 80.97 | 85.36 | 71.55 | 84.58 | 84.70 |
| OFA$_{Base}$ (Wang et al., 2022c) | 88.48 | 90.67 | 83.30 | 81.39 | **87.15** | 74.29 | 82.29 | 82.31 |
| OFA$_{Huge}$ | **92.04** | **94.03** | **88.44** | **87.86** | **91.70** | **80.71** | **88.07** | **88.78** |
| **UnIVAL** (ours) | **89.12** | **91.53** | **85.16** | **82.18** | 86.92 | **75.27** | **84.70** | **85.16** |

Table 5: **Finetuning for Visual Grounding on RefCOCO, RefCOCO+, and RefCOCOg datasets. UnIVAL** achieves the new SoTA results among comparable model sizes. We report Acc@0.5.

**Visual Grounding.** We evaluate the ability of the model to localise spatially the text in the image. This task consists of predicting the coordinates of bounding box given an input text. The task is cast as sequence generation task, where the model outputs a sequence of 4 pixel locations corresponding to the 4 corners of the bounding box. Tab.5 shows that we achive new SoTA results on all 3 benchmarks. Interestingly, our scores are better than the reported OFA scores, which additionally pretrain for object detection.

| Model | VQAv2 | | SNLI-VE | |
|---|---|---|---|---|
| | test-dev | test-std | dev | test |
| UNITER (Chen et al., 2020c) | 73.8 | 74.0 | 79.4 | 79.4 |
| OSCAR (Li et al., 2020b) | 73.6 | 73.8 | - | - |
| VILLA (Gan et al., 2020) | 74.7 | 74.9 | 80.2 | 80.0 |
| VinVL (Zhang et al., 2021) | 76.5 | 76.6 | - | - |
| UNIMO (Li et al., 2020a) | 75.0 | 75.3 | 81.1 | 80.6 |
| ALBEF (Li et al., 2021a) | 75.8 | 76.0 | 80.8 | 80.9 |
| ViCHA (Shukor et al., 2022) | 75.0 | 75.1 | 79.9 | 79.4 |
| METER (Dou et al., 2021) | 77.7 | 77.6 | 80.9 | 81.2 |
| *Text-generation approaches* | | | | |
| VL-T5 (Cho et al., 2021) | - | 70.3 | - | - |
| UniTAB (Yang et al., 2021b) | 70.7 | 71.0 | - | - |
| GIT-L (Wang et al., 2022a) | 75.5 | - | - | - |
| OmniVL (Wang et al., 2022b) | 78.3 | 78.4 | - | - |
| OFA$^{\dagger}_{Base}$ (Wang et al., 2022c) | 77.0 | 77.1 | 78.8 | 78.6 |
| *Large-scale pretraining* | | | | |
| SimVLM$_{Large}$ (Wang et al., 2021) | 79.3 | 79.6 | 85.7 | 85.6 |
| Florence (Yuan et al., 2021) | 80.2 | 80.4 | - | - |
| PaLM-E 84B (Driess et al., 2023) | 80.5 | – | - | - |
| **UnIVAL** (ours) | 77.0 | 77.1 | 78.2 | 78.6 |

| Model | Cross-Entropy Optimization | | | |
|---|---|---|---|---|
| | BLEU@4 | METEOR | CIDEr | SPICE |
| VL-T5 (Cho et al., 2021) | 34.5 | 28.7 | 116.5 | 21.9 |
| OSCAR (Li et al., 2020b) | 37.4 | 30.7 | 127.8 | 23.5 |
| UniTAB (Yang et al., 2021b) | 36.1 | 28.6 | 119.8 | 21.7 |
| VinVL (Zhang et al., 2021) | 38.5 | 30.4 | 130.8 | 23.4 |
| UNIMO (Li et al., 2020a) | 39.6 | - | 127.7 | - |
| GIT-L (Wang et al., 2022a) | 42.0 | 30.8 | 138.5 | 23.8 |
| OmniVL (Wang et al., 2022b) | 39.8 | - | 133.9 | - |
| OFA$^{\dagger}_{Base}$ (Wang et al., 2022c) | 42.5 | 30.6 | 138.1 | 23.7 |
| *Large-scale pretraining* | | | | |
| LEMON (Hu et al., 2022) | 41.5 | 30.8 | 139.1 | 24.1 |
| SimVLM$_{Large}$ (Wang et al., 2021) | 40.3 | 33.4 | 142.6 | 24.7 |
| PaLM-E 84B (Driess et al., 2023) | – | – | 138.0 | – |
| **UnIVAL** (ours) | 42.0 | 30.5 | 137.0 | 23.6 |

Table 6: **Finetuning on Image-Text understanding and generation tasks such as VQAv2, SNLI-VE and Image Captioning.** Our text-generation based approach is competitive with other SoTA, while using less pretraining data.

**Multimodal understanding tasks.** We evaluate on VQA and Visual entailment tasks, that we cast as text generation. Tab.6 shows a comparison with other approaches. Despite pretraining on less data for less number of steps, our approach is on par with the previous unified model OFA (Wang et al., 2022c) finetuned from the author's released checkpoint ($OFA^{\dagger}_{Base}$). For comparable scale, we significantly outperform $GIT_L$ (Wang et al., 2022a) that uses CLIP-ViT-L as image encoder. Our model is competitive with other SoTA models trained on large datasets that casted the task as classification. Note that, we evaluate both our model

and OFA, with beam search for VQA, instead of all-candidate evaluation. For SNLI-VE, our approach uses only the image and the text hypothesis, without the text premise as previously done in OFA (Wang et al., 2022c). The results on SNLI-VE suggest that unified models such OFA and our models underperform on the visual entailment task.

**Multimodal generation tasks.** We evaluate the model for image captioning on COCO dataset (Lin et al., 2014), and report the scores on the Karpathy test split. Tab.6 shows that we are comparable with OFA. Compared to the previous OmniVL model (Wang et al., 2022b) that pretrain on both image and video text datasets, we largely outperform it by more than 3 points CIDEr. Our model is very close to other SoTA such as GIT-L and large-scale trained ones such as LEMON and PaLM-E 84B.

### 4.1.2 Video-Text tasks

Here we evaluate the model on different video-text tasks.

| Method | #PT images/videos | MSRVTT-QA | MSVD-QA |
|---|---|---|---|
| ClipBERT (Lei et al., 2021) | 0.15M/- | 37.4 | - |
| JustAsk (Yang et al., 2021a) | -/69M | 41.5 | 46.3 |
| ALPRO (Li et al., 2022a) | 3M/2.5M | 42.1 | 45.9 |
| MERLOT (Zellers et al., 2021) | -/180M | 43.1 | - |
| VIOLET (Fu et al., 2021) | 3.3M/182M | 43.9 | 47.9 |
| All-in-one (Wang et al., 2023a) | -/283M | 46.8 | 48.3 |
| GIT (Wang et al., 2022a) | 800M/- | 43.2 | 56.8 |
| OmniVL (Wang et al., 2022b) | 14M/2.8M | 44.1 | 51.0 |
| LAVENDER (Li et al., 2022c) | 14M/14.4M | 45.0 | 56.6 |
| UnIVAL (ours) | 14M/2.5M | 43.48 | 49.55 |

Table 7: **Finetuning for VideoQA on MSRVTT-QA and MSVD-QA datasets**. The text-generation based **UnIVAL** model is competitive with SoTA models customized for videos or trained on significantly larger datasets. We report the accuracy.

**Video question answering.** We evaluate for VideoQA on MSRVTT-QA and MSVD-QA (Xu et al., 2017) datasets. Tab.7 shows a comparison with other approaches. On MSRVTT-QA, we outperform large scale pretrained models like GIT, including models trained on more videos (MERLOT) and customised for VideoQA (JustAsk). We are competitive with the unified video model LAVENDER with heavier vision encoder (Video Swin), trained on more videos (and restrict the generated answers to one word), and the ununified OmniVL targeting both images and videos. On MSVD-QA, we have competitive performance to previous work.

| Method | #PT Image (Video) Data | MSRVTT B@4 | M | R | C |
|---|---|---|---|---|---|
| UniVL (Luo et al., 2020) | (136M) | 42.2 | 28.2 | 61.2 | 49.9 |
| SwinBERT (Lin et al., 2022) | - | 41.9 | 29.9 | 62.1 | 53.8 |
| CLIP4Caption (Tang et al., 2021) | - | 46.1 | 30.7 | 63.7 | 57.7 |
| MV-GPT[T] (Seo et al., 2022) | (53M) | 48.9 | 38.7 | 64.0 | 60.0 |
| LAVENDER (Li et al., 2022c) | 14M (14.4M) | - | - | - | 60.1 |
| UnIVAL (ours) | 14M (2.5M) | 46.42 | 29.01 | 62.92 | 60.5 |

| Method | ActivityNet-Captions B@3 | B@4 | M |
|---|---|---|---|
| DCEV (Krishna et al., 2017a) | 4.09 | 1.60 | 8.88 |
| DVC (Li et al., 2018) | 4.51 | 1.71 | 9.31 |
| Bi-SST (Wang et al., 2018a) | – | – | 10.89 |
| HACA (Wang et al., 2018b) | 5.76 | 2.71 | 11.16 |
| MWSDEC (Rahman et al., 2019) | 3.04 | 1.46 | 7.23 |
| MDVC (Iashin & Rahtu, 2020b) | – | 1.46 | 7.23 |
| BMT (Iashin & Rahtu, 2020a) | 4.63 | 1.99 | 10.90 |
| MV-GPT[T] (Seo et al., 2022) | – | 6.84 | 12.31 |
| UnIVAL (ours) | 7.67 | 4.76 | 10.51 |

Table 8: **Finetuning for Video Captioning on MSRVTT and ActivityNet-Captions. UnIVAL** is competitive with other task/modality-customized SoTA that are trained on larger datasets. [T]: uses in addition text transcript. For ActivityNet-Captions we use ground-truth action proposals.

**Video captioning.** We evaluate our model for Video Captioning. Tab.8 shows that our model is very competitive with other approaches customized for videos, trained on much larger datasets (LAVENDER) and use speech transcript as additional input (MV-GPT). On ActivityNet-Caption with ground truth proposal, we outperform previous approaches by significant margin as per the B@4 metric and we are competitive with the current SoTA MV-GPT.

| Dataset | Method | BLEU$_1$ | BLEU$_2$ | METEOR | CIDEr | SPICE |
|---|---|---|---|---|---|---|
| Audiocaps | (Kim et al., 2019b) | 0.614 | 0.446 | 0.203 | 0.593 | 0.144 |
| | (Xu et al., 2021) | 0.655 | 0.476 | 0.229 | 0.660 | 0.168 |
| | (Mei et al., 2021) | 0.647 | 0.488 | 0.222 | 0.679 | 0.160 |
| | (Liu et al., 2022) | 0.671 | 0.498 | 0.232 | 0.667 | 0.172 |
| | **UnIVAL** (ours) | **0.690** | **0.515** | **0.237** | **0.713** | **0.178** |
| Clotho v1 | (Takeuchi et al., 2020) | 0.512 | 0.325 | 0.145 | 0.290 | 0.089 |
| | (Koizumi et al., 2020) | 0.521 | 0.309 | 0.149 | 0.258 | 0.097 |
| | (Chen et al., 2020a) | 0.534 | 0.343 | 0.160 | 0.346 | 0.108 |
| | (Xu et al., 2020) | 0.561 | 0.341 | 0.162 | 0.338 | 0.108 |
| | (Eren & Sert, 2020) | **0.590** | 0.350 | **0.220** | 0.280 | - |
| | (Xu et al., 2021) | 0.556 | 0.363 | 0.169 | 0.377 | **0.115** |
| | (Koh et al., 2022) | 0.551 | **0.369** | 0.165 | **0.380** | 0.111 |
| | **UnIVAL** (ours) | 0.569 | 0.367 | 0.178 | **0.380** | 0.114 |

Table 9: **Finetuning on the new audio-text modality for audio-captioning.** We compare **UnIVAL** to other audio-text models on Audiocaps and Clotho v1 datasets. Despite not using audio-text during pretraining **UnIVAL** is very competitive with other customized SoTA. We compare with models that rely only on audio as input. The best and next best scores are **bolded** and underlined respectively.

### 4.1.3 Audio-Text Tasks

In addition to considering only modalities seen during pretraining, we explore if **UnIVAL** also works well for potential new ones. To this end, we evaluate the model on the new audio modality. We use an additional audio encoder pretrained on audio classification and finetune jointly the encoder and the core model pretrained on our image/video-text data.

**Audio captioning.** We evaluate the model on standard audio captioning datasets; Clotho v1 and Audiocaps. Tab.9 shows a comparison with other approaches that take solely the audio as input. Interestingly, we significantly outperform other approaches on Audiocaps, and we are competitive with the current SoTA on the small Clotho v1 dataset.

## 4.2 Evaluation without finetuning

| Model | VQAv2 test-dev Acc | COCO Caption Val/Test CIDEr | RefCOCO+ Val Acc@0.5 |
|---|---|---|---|
| Unified-IO$_{Base}$ (Lu et al., 2022a) | 61.8 | 104.0/– | – |
| OFA$_{Base}$ (Wang et al., 2022c) | 68.91 | 74.47/75.27 | 30.45 |
| **UnIVAL** | 70.18 | 90.07/91.04 | 70.81 |

Table 10: **Evaluation without finetuning. UnIVAL** outperforms OFA and competitive with Unified-IO trained on more data.

| Model | OKVQA Val Acc | VizWiz Val Acc | NoCaps CIDEr (out-domain) | MSRVTT-QA Test Acc | MSVD-QA Test Acc |
|---|---|---|---|---|---|
| Unified-IO$_{Base}$ (Lu et al., 2022a) | 37.8 | 45.8 | – | – | – |
| OFA$_{Base}$ (Wang et al., 2022c) | 40.16 | 17.33 | 48.95 | – | – |
| LAVENDER (Li et al., 2022c) | – | – | – | 4.5 | 11.6 |
| Flamingo-3B (Alayrac et al., 2022) | 41.2 | 28.9 | – | 11.0 | 27.5 |
| **UnIVAL** | 38.91 | 20.22 | 47.68 | 5.84 | 21.15 |

Table 11: **Zero-Shot Evaluation.** Scores in gray means the dataset is used during pretraining. **UnIVAL** is competitive with modality-specific models.

**Evaluation on seen datasets.** Following Lu et al. (2022a), we directly evaluate the representation learned during pretraining without task-specific finetuning. We compare our model to different baselines following the same setup, with the main difference that other baselines pretrain longer, on significantly larger datasets and more tasks. Tab.10 shows that our approach outperforms the most similar baseline OFA on all tasks. Compared to Unified-IO, we are significantly better on VQAv2, despite pretraining on less VQA datasets.

**Evaluation on unseen datasets (zero-shot).** We follow the same previous setup, but we evaluate the model on new datasets, unseen during pretraining. Tab.11 shows a comparison with other models on several image and video-text datasets. Our model is very competitive to OFA, and close to Unified-IO (grayed scores) on OKVQA. However, Unified-IO pretrains on both OKVQA and VizWiz. Compared to the unified video-language model LAVENDER, we significantly outperform it on video tasks. Our approach attains close performance to the large-scale Flamingo-3B model on OKVQA and MSVD-QA.

### 4.3 Generalization to new tasks and modalities

In this section we investigate the importance of pretraining on different modalities for the generalization to new tasks and modalities. This is important in scenarios where we we want the model to do well also on potentially novel tasks or modalities. Specifically, we want to validate the following hypothesis; *pretraining on more modalities, and thus on more tasks, allows to learn more modality and task-agnostic representation.*

| Modality | Multitask | Audiocaps |
|---|---|---|
| Image-Text | ✗ | 54.4 |
| Image-Text | ✓ | **57.6** |
| Text | ✗ | 53.2 |
| Image-Text | ✓ | 58.4 |
| Video-Text | ✓ | 57.4 |
| Image-Text+Video-Text | ✓ | **58.8** |

Table 12: **Finetuning for Audio Captioning on the Audiocaps dataset.** We compare different initialization (after pretraining on Images-Text (I), Videos-Text (V), or Text (T)) for audio captioning. Pretraining on more modalities leads to better results when finetuning on audio captioning, a task not seen during pretraining.

| Method | CLIP score ↑ |
|---|---|
| Text | 31.0 |
| Image-Text | **31.6** |
| Image-Text+Video-Text | 31.3 |

Table 13: **Finetuning for text-to-image generation on COCO dataset**. Multimodal pretraining improves the results when finetuning on new text-to-image generation, a task not seen during pretraining.

**Better initialization for new modalities: from vision-language to audio-language tasks.** We finetune our model for audio captioning on the Audiocaps dataset. To compare the effect of pretraining on more tasks and modalities, we evaluate the same model with different initialization; pretraining on text (the model initialized from BART), pretraining on image-text (with and without multitask pretraining), pretraining on video-text and pretraining on both image and video-text. We pretrain for the same number of epochs. Tab.12 shows that pretraining on image-text and video-text data leads to better scores on Audiocaps, compared to the model pretrained on text. Interestingly, the model pretrained on both modalities attain the best scores. This support our underlying hypothesis. We also show the importance of multitask pretraining, by comparing two models trained on image-text tasks; one with single task on CC3M and CC12M (12.8M examples) and another one with multitask on COCO, VG, SBU, CC3M, VQAv2 and RefCOCO+ (5.6M examples). The results validates again the importance of multitasking in generalization to new modalities/tasks.

**Better initialization for new tasks: from multimodal input to multimodal output.** Here, we investigate if our pretrained model can be a good initialization to add new tasks. We experiment with a more challenging scenario; text-to-image generation. We finetune the model with different initialization on the COCO dataset and report the CLIP score (Wu et al., 2022). Tab.13 shows that pretraining on either image-text or video-text data helps to get additional improvement, with more improvement coming from pretraining on image-text tasks.

## 5 Weight Interpolation of UnIVAL Models

Previously, we showed the synergy between tasks and modalities that results from multitask pretraining. Here, instead, we use interpolation in the weight space to leverage this synergy. We follow the literature on weight interpolation (Izmailov et al., 2018; Wortsman et al., 2022; Rame et al., 2022) to merge models finetuned on different multimodal tasks, without inference overhead. Our framework is an ideal candidate for this investigation, due to the unified architecture and the shared pretraining (Neyshabur et al., 2020), which naturally enforces linear mode connectivity (Frankle et al., 2020) and thus averegability (without requiring weight permutation as in Ainsworth et al. (2022)) across finetuned weights. We consider 4 image-text tasks; Image Captioning (IC), VQA, Visual Grounding (VGround) and Visual Entailment (VE), and provide similar results for video tasks in Appendix J. Then, given two models with weights $W_1$ and $W_2$ finetuned on 2 different tasks among those 4 image-text tasks, we analyze the performance of a new model whose weights are $W = \lambda \cdot W_1 + (1 - \lambda) \cdot W_2$, where $\lambda \in [0, 1]$. We propose to study the following questions.

***Scenario 1*: can we trade off between 2 different multimodal tasks by weight interpolation?** In Fig.2 we plot the performances of the model whose weights are defined by interpolation ($\lambda \in [0,1]$) between different finetuned weights. All weights were initialized from the same "pretrain init", which performs badly on the considered tasks (grey star). By vanilla finetuning (blue stars at the endpoints of the blue line) on a target task, we consistently improve results on the corresponding metric, yet at the cost of severe performance degradation on other tasks; this suggests that the different tasks are in tension, and optimizing one degrades another. Then, by weight interpolation between these two vanilla finetuned endpoints, we reveal a convex front of solutions that trade-off between the different abilities, validating that *we can effectively combine the skills of expert models finetuned on diverse multimodal tasks*. Actually, it is even possible to find an interpolating coefficient $\lambda$ so that the interpolated model outperforms the specialized one: *e.g.*, in the first subplot, the CIDEr score of the model for $\lambda = 0.8$ with weights $0.8 \cdot \theta_{Cap} + 0.2 \cdot \theta_{VQA}$ is $138.26$ vs. $136.10$ for the captioning expert $\theta_{Cap}$ corresponding to $\lambda = 1$. We speculate the interpolated model benefits from the synergy between different tasks.

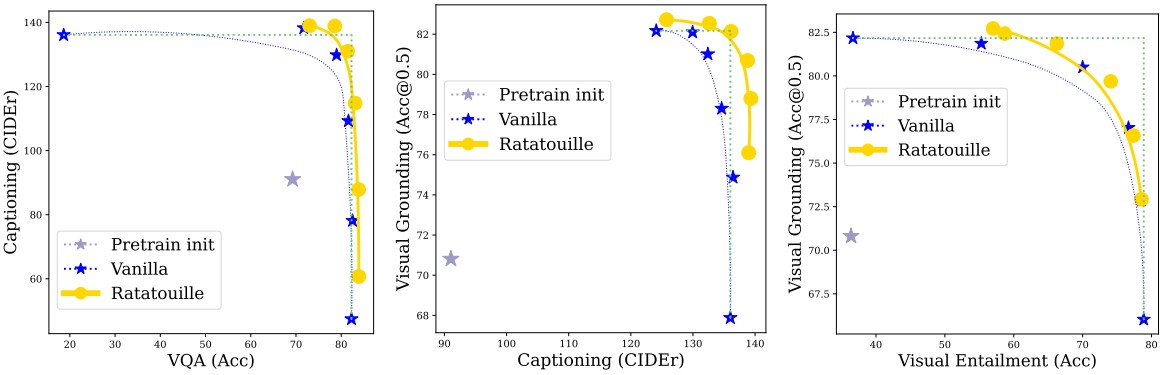

Figure 2: **Weight interpolation between models trained on different multimodal tasks.**

Besides, we also experiment a more costly finetuning approach, ratatouille (Rame et al., 2023a), where each finetuning leverages the other tasks as auxiliary tasks. For example, when considering VQA as the target task; (i) we first finetune weights on the auxiliary tasks (VGround, VE and IC); (ii) then we launch multiple finetunings on VQA from these auxiliary weights; (iii) we uniformly average all the weights finetuned on VQA to obtain $W_{VQA}^r$. Similarly, to obtain $W_{IC}^r$ specialized on IC, we apply the same recipe but this time the final finetunings are on the target task IC. Then, as shown on the left subplot from Fig.2, we plot the performances for $W^r = \lambda \cdot W_{VQA}^r + (1 - \lambda) \cdot W_{IC}^r$ where $\lambda \in [0,1]$. The obtained (yellow) front is to the right and above the vanilla (blue) front, validating the superior performances of ratatouille.

***Scenario 2*: given $N$ models trained on different multimodal tasks, can we obtain a single model that is good on seen, and new unseen tasks?** We consider $N = 4$ models with weights $\{W_i\}_{i=1}^N$ finetuned independently, one for each task. This time, we simply average them uniformly: $\lambda = 1/N$, and then we consider the weight average (WA) $\frac{1}{N} \sum_{i=1}^N W_i$ and plot its performance in Fig.3. First, we observe that this vanilla WA outperforms the "pretrain init", evaluated on training tasks (in-distribution or ID setting) but also when evaluated on new tasks (out-of-distribution or OOD setting), without any additional training. Performances are improved when averaging uniformly ratatouille finetunings $\frac{1}{N} \sum_{i=1}^N W_i^r$. We also consider the fusing (Choshen et al., 2022) strategy, which considers the average the auxiliary weights $\frac{1}{N} \sum_{i=1}^N W_i$ as the initialization for a second step of finetunings on the target tasks to obtain $\{W_i^f\}_{i=1}^N$, and then report the performance for $\frac{1}{N} \sum_{i=1}^N W_i^f$; fusing performs better than vanilla fine-tuning in ID, but not OOD. When comparing fusing and ratatouille, they perform similarly ID; yet, in the OOD setting, ratatouille outperforms fusing, validating the results from Rame et al. (2023a). In conclusion, these experiments show that uniform averaging can merge different finetuned models to get one general model that performs well on all seen and unseen tasks. Though, results could be improved with tailored selection of the interpolation coefficients $\lambda$.

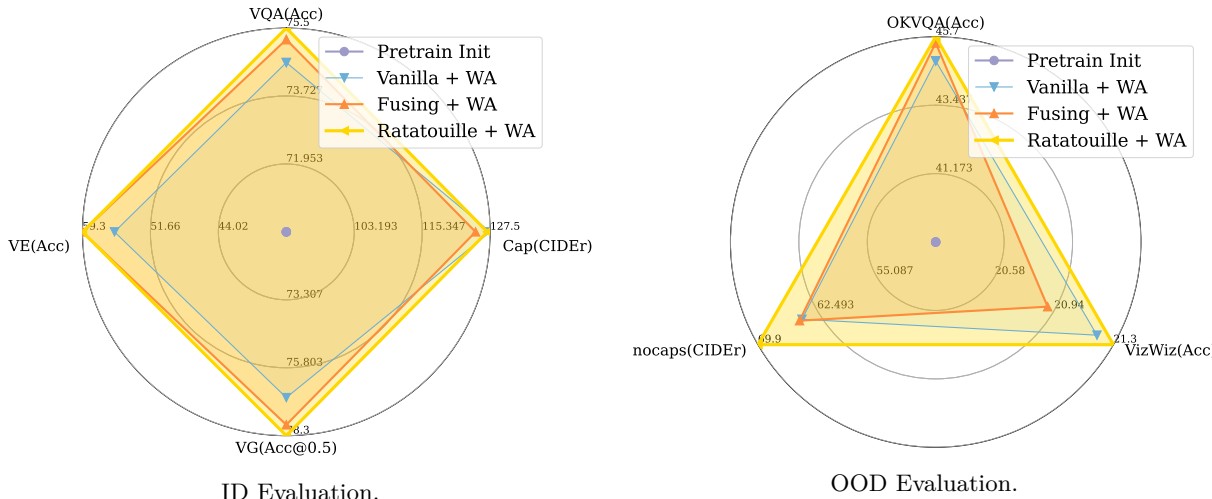

ID Evaluation.

OOD Evaluation.

Figure 3: **Finetuning for OOD.** We uniformly average the models finetuned on 4 image-text tasks and evaluate the resulting model on the same (ID) and new (OOD) tasks.

## 6 Discussion

**Limitations and discussion.** Despite the good quantitative results, we find that UnIVAL suffers from several limitations. First, UnIVAL can **hallucinate**; describe objects not present in the image (Rohrbach et al., 2018), prioritizing coherence in its generation rather than factuality. In the case of VQA, the model can generate plausible response that can not answered given the image. A similar challenge arises in visual grounding, where UnIVAL may ground objects that are not mentioned in the text or not present in the image. Nonetheless, in comparison to other large models like Flamingo (Alayrac et al., 2022), we show in Appendix K that UnIVAL demonstrates reduced hallucinations. The reason why models such as Flamingo hallucinate more might be due to using frozen LLMs, a component that is known to be susceptible to hallucinate (Zhang et al., 2023; Shukor et al., 2023b). Second, it struggles in **complex instruction following.** We have observed that the model's performance is suboptimal to intricate instructions, such as identifying a specific object in the presence of similar alternatives, detecting small or distant objects, and recognizing numerals. Other limitations that are also important to address are; social and other biases that can be learned from the datasets, toxic generations, and explainable generation. These limitations might not be solved by merely scaling the model, and might need different approaches Shukor et al. (2023b); Rame et al. (2023b). In Appendix K, we provide a detailed discussion on these limitations and interesting future directions.

**Conclusion.** In this study, we introduce UnIVAL, the first unified model capable of supporting image, video, and audio-text tasks. We achieve this with a relatively small model with $\sim 0.25B$ parameter on dataset of relatively small sizes. Our unified system, pretrained with multitasking, offers several advantages. It harnesses the synergies between diverse tasks and modalities, enables more data-efficient training, and exhibits strong generalization capabilities to novel modalities and tasks. The unification aspect of our strategy paves the way to interesting techniques to merge models finetuned on different multimodal tasks: we demonstrate that, in addition to multitask pretraining, merging by weight interpolation can further exploit the tasks diversity. Ultimately, we aspire that our work inspires the research community and accelerates the progress toward constructing modality-agnostic generalist assistant agents.

## 7 Acknowledgments

This work was supprted by HPC resources of CINES and GENCI. The authors would like to thank the staff of CINES for technical support in managing the Adastra GPU cluster, in particular; Jean-Christophe Penalva, Johanne Charpentier, Mathieu Cloirec, Jerome Castaings, Gérard Vernou, Bertrand Cirou and José Ricardo Kouakou. This work was also partly supported by ANR grant VISA DEEP (ANR-20-CHIA-0022).

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

## Appendix

The Appendix is organized as follows:

- Section A: model card.

- Section B: detailed discussion about related work.

- Section C: background on unified models and different unification axes.

- Section D: details about model architecture.

- Section E: image and video-text pretraining tasks.

- Section F: illustration and details about multimodal curriculum learning.

- Section G: datasets and implementation details.

- Section H: finetuning only the linear connection (Parameter-Efficient Finetuning).

- Section I: ablation study including knowledge transfer across modalities and training efficiency.

- Section J: additional quantitative results.

- Section K: discussion of several limitations and future directions.

- Section L: qualitative results of several image-text tasks.

## A    Model Card

In the following table, we detail our model card (Mitchell et al., 2019).

| Model Details | |
|---|---|
| Model Date | July 2023 |
| Model Type | Transformer encoder-decoder pretrained on text and trained end-to-end to be conditioned on image, video and audio input. Modality-specific encoders are based on convnets and pretrained from classification on public benchmarks. All input tokens are concatenated and fed to the encoder. The text generation is conditioned on other modalities via cross-attention. (See Section for details.) |
| **Intended Uses** | |
| Primary Intended Uses | The primary use is research on unified multimodal models that span a wide range of applications such as; image/video/audio captioning, image/video question answering, grounding/detection and image generation. In addition, the study of the limitation and biases of such kind of model, and novel approach for efficient training and adaptation. Other similar multimodal applications can also be considered, like multimodal dialogue, and text-guided robotics applications. |
| Primary Intended Users | The research community. The model will be made public. |

| Out-of-Scope Uses | Any downstream applications that can cause harm to society, or without mitigation of associative safety measures. |
|---|---|

### Factors

| Card Prompts – Relevant Factor | The model is trained on english and based on BART (Lewis et al., 2020) language model. The model should not be used any downstream application without propoer factor analysis. |
|---|---|
| Card Prompts – Evaluation Factors | The model inherits the biases and risks of the pretrained language model (Lewis et al., 2020). It may also hallucinates some information not present in the conditioned modality. On some tasks we constraints the text generation to predifined set of answers, however, generally, there is no mechanism that force it to not produce toxic or racist output on all tasks. |

### Metrics

| Model Performance Measures | The performance using standard metrics to evaluate the model performance on several public benchmarks, such as; Visual Question Answering (accuracy on VQAv2, OKVQA, ,VizWiz, MSVD-QA and MSRVTT-QA), Visual Grounding (IoU>0.5 on RefCOCO, RefCOCO+ and RefCOCOg), Image Captioning (CIDEr, METEOR, BLEU, SPICE on MSCOCO, MSR-VTT, Audiocaps and Clotho v1) and Text to Image Generation (CLIP score on MSCOCO). |
|---|---|
| Decision thresholds | N/A |
| Approaches to Uncertainty and Variability | The relatively costly pretraining prevent from doing several runs, however the different ablation study and the evaluation on many datasets validate the overall performance of the model. |

### Evaluation Data

| Datasets | Check Tab. 15 for more details. |
|---|---|
| Motivation | The datasets span different standard benchamrks across image, video and audio modalities. This show the overall capability of the model to process different modalities. |
| Preprocessing | Text is process with BPE tokenizers, audio is transformer to mel spectorgram and we randomly sample some frames from videos. Some addition data augmentation techniques are used during training. |

### Training Data

| Datasets | We only use public datasets, such as image captioning (COCO (Lin et al., 2014), Visual Genome (VG) (Krishna et al., 2017b), SBU (Ordonez et al., 2011), CC3M (Sharma et al., 2018) and CC12M (Changpinyo et al., 2021) (only in the first stage)), VQA (VQAv2 (Goyal et al., 2017), GQA (Hudson & Manning, 2019), VG (Krishna et al., 2017b)), Visual Grounding (VGround) and referring expression comprehension (RefCOCO, RefCOCO+, Ref-COCOg (Yu et al., 2016)), video captioning (WebVid2M (Bain et al., 2021)) and video question answering (WebVidQA (Yang et al., 2021a)). We only use the training sets during pretraining. |
|---|---|
| **Quantitative Analyses** | |
| Unitary Results | Our unified model is competitive to state of the art approaches customized for less modalities. It attains state of the art results on Visual Grounding and Audio Captioning. Please check Sec.4.1 for more details. |
| Intersectional Results | N/A. |
| **Ethical Considerations** | |
| Data | We use only public benchmarks, however some benchmarks are not filtered from racist, sexist or otherwise harmful content. |
| Human Life | The model is not intended to be used for safety critical applications. |
| Mitigations | Constrained text generation can be adapted for some tasks. However, for open-ended generation post processing or some engineered prompts might mitigate some of the biases. Overall, filtering the pretraining data can be ver effective approach. |
| Risks and Harms | We use public datasets. Not all of them are filtered from from toxic and personal data. |
| Use Cases | Forcing the model (finetuning or prompting) to generate harmful or racist text. Other use cases regarding general language models are also relevant. |

Table 14: **UnIVAL Model Card.** We follow the framework of (Mitchell et al., 2019).

# B  Related Work

**Unimodal pretraining**  Pretraining on large uncurated datasets has been a substantial ingredients in the vision and NLP communities to develop powerful models that generalize to a wide range of tasks. For vision models, supervised (Touvron et al., 2021; Dehghani et al., 2023) and self supervised (Chen et al., 2020b; Caron et al., 2020; Zbontar et al., 2021; He et al., 2022) techniques have extensively investigated , while for NLP, the widely used training objective is next token prediction (Brown et al., 2020; Hoffmann et al., 2022; Touvron et al., 2023).

Recently, these domains started to converge on a simple training paradigm; joint scaling of the pretraining data, model size and compute, while using a unified architecture and training objective. Surpassing a certain scaling threshold has elicited new emergent capabilities, especially in LLMs (Brown et al., 2020; Chowdhery et al., 2022), that allows such models to solve new reasoning tasks that were out of reach few years ago. Once

such models are available, they can be seamlessly adapted without retraining, via prompting such zero-shot or few-shot In Context Learning. Scaling vision transformer models (Dehghani et al., 2023) lead to be more robust and aligned to human object recognition.

While being very successful, training such models is hard, extremely costly and need dedicated infrastructure. However, the public release of many of these models allow to leverage them for variety of tasks. In this work we leverage unimodal pretrained models for multimodal tasks.

**Multimodal pretraining.**   So far, most of the effort to build multimodal models have been focused on vision-language pretraining. Contrastive based approaches (Radford et al., 2021; Jia et al., 2021) try to learn shared and aligned latent space by training on hundred of millions of data. More data efficient approaches (Shukor et al., 2022; Li et al., 2021a; 2022b; Dou et al., 2021; Singh et al., 2022), have relied on additional multimodal interaction modules and variety of training objectives such as image-text matching, masked language modeling and image-text contrastive (Chen et al., 2020c; Kim et al., 2021; Lu et al., 2019; Zhang et al., 2021). In the video-language community, similar approaches have been mildly adapted to model the interaction between language and frames sequences (Cheng et al., 2022; Wang et al., 2023a; Fu et al., 2021; Zellers et al., 2021; Yang et al., 2021a). Few work have targeted both image and video language pretraining (Wang et al., 2022b).

These works have been following the scaling trend as in unimodal pretraining. Scaling the model went from couple of billions of parameters (Yu et al., 2022; Wang et al., 2022e;a) to tens of billions (Chen et al., 2022b; Alayrac et al., 2022).

**Unified models**   Building unified systems has been triggered first in the NLP community. (Raffel et al., 2020) proposed the T5 transformer model, a text-to-text framework, where the same pretrained model is used to solve many NLP tasks, each one is described by task-specific textual prefix. Since then, building general textual models has been heavily investigated by LLMs (Brown et al., 2020; Rae et al., 2021; Chowdhery et al., 2022). The success of unified Language models, have inspired other communities. In the vision community, (Chen et al., 2022a) proposed a pixel-to-sequence framework to unify different vision tasks such as object detection and instance segmentation. For multimodal tasks, (Cho et al., 2021) proposed to unify vision-language tasks, including discriminative ones, as conditional text generation. This was followed by (Yang et al., 2021b), which targets also grounded tasks and does not rely on an object detection model. OFA (Wang et al., 2022c) then proposed a large scale sequence-to-sequence framework, and extended previous approaches to more image-text tasks, including text to image generation. Similarly, Unified-IO (Lu et al., 2022a), in addition to image-text tasks, targets many visual tasks including dense prediction such as depth estimation and image segmentation. The most closest to us is the work of OFA and Unified-IO, however, we propose to unify tasks across many modalities, and use smaller model and dataset sizes.

**Efficient multimodal learning**   The current paradigm in training multimodal models is to train all model parameters, even when using pretrained models (Chen et al., 2022b; Wang et al., 2022c; Li et al., 2022b). Despite attaining SoTA, these approaches are extremely costly to train. To overcome this, recent approaches showed that pretrained models, generalize well to multimodal tasks, where it is possible to use a frozen LM with a powerful multimodal encoder such as CLIP, and train only a handful of parameters, such as the vision encoder (Eichenberg et al., 2021), the vision connector (Merullo et al., 2022; Mañas et al., 2022; Koh et al., 2023; Li et al., 2023) or additionally the Adapters (Eichenberg et al., 2021; Yang et al., 2022). This paradigm was then generalized in (Shukor et al., 2023a), to other modalities, such video and audio, where the authors showed that it is even possible train only a linear projection layer to adapt pretrained unimodal encoder (*e.g.*, pretrained on ImageNet) and a language decoder to do multimodal tasks.

Another line of research, is data-efficient approaches, recent work shows that it is possible to get comparable results by training on significantly less data, by designing better training objectives (Shukor et al., 2022), data augmentation (Li et al., 2021b) and curriculum learning (Srinivasan et al., 2022). In this work, we focus on parameter-efficient finetuning, especially, training only the linear connection.

**Weight interpolation and mutltimodal tasks.**   Our strategy enable the training of multiple expert models with diverse specializations. To combine them, we leverage a simple yet practical strategy: *linear*

*interpolation in the weight space*, despite the non-linearities in the network's architecture. This weight averaging (WA) strategy is in line with recent findings on linear mode connectivity (Frankle et al., 2020; Neyshabur et al., 2020, LMC): weights fine-tuned from a shared pre-trained initialization remain linearly connected. Recent works (Ilharco et al., 2023; Daheim et al., 2023; Ortiz-Jimenez et al., 2023) suggest that averaging networks in weights can combine their abilities without any computational overhead; for instance, the average of an English summarizer and an English-to-French translator will behave as a French summarizer (Jang et al., 2023). Model soups approaches (Wortsman et al., 2022; Rame et al., 2022) improve out-of-distribution generalization and show that weight averaging actually approximates predictions averaging (Lakshminarayanan et al., 2017) when the LMC holds. The LMC was extended to weights fine-tuned with different losses (Rame et al., 2022; Croce et al., 2023; Rame et al., 2023b) or on different datasets (Matena & Raffel, 2022; Ilharco et al., 2022; Choshen et al., 2022; Don-Yehiya et al., 2023; Rame et al., 2023a; **?**). Moreover, several other merging approaches (Matena & Raffel, 2022; Yadav et al., 2023) have been proposed, though with arguably minor empirical gains over the simpler linear interpolation. For example, (Matena & Raffel, 2022) considers the Fisher information; (Yadav et al., 2023) resolve updates conflicts across weights. Neuron permutations strategies (Entezari et al., 2022; Ainsworth et al., 2022; Jordan et al., 2023) address the ambitious challenge of enforcing connectivity across weights with different random initializations, though so far with moderate empirical results. Most of exisiting WA approaches consider very similar tasks, such as image classifications from different datasets or text classification/generation. Interpolating weights of models finetuned on different multimodal tasks, is little investigated, with no work exploring this technique in multimodal foundation models. The most similar and concurrent work is the recent Sung et al. (2023) applying a complex architecture-specific merging strategy involving weight averaging for models pretrained on different modalities. Another difference, is that we explore WA for multimodal downstream tasks.

## C   Unified Foundation Models: 4 Unification axes

While many previous works have attempted to build unified models, they still have some customization in terms of architectures and tasks. Our work tries to unify most aspects of the model, following a recent line of work (Wang et al., 2022c). In the following, we detail the 4 unification axes that distinguish our work from previous ones.

**Unified input/output.**   To have a unified model, it is important to have the same input and output format across all tasks and modalities. The common approach is to cast everything to sequence of tokens as in language models. Multimodal inputs, such as images, videos and audios can be transformed to tokens by patchifying or using shallow modality-specific projections. Multimodal outputs can also be discritized, by using VQ-GAN for images and discrete pixel locations for visual grounding. A unified vocabulary is used when training the model.

**Unified model.**   The unified input/output representation allows to use a single model to solve all tasks, without the need to any adaptation when transitioning from the pretraining to the finetuning phase (*e.g.*, no need for task-specific heads). In addition, the current advances in LLMs, especially their generalization to new tasks, make it a good choice to leverage these models to solve multimodal tasks. The common approach is to have a language model as the core model, with light-weight modality-specific input projections.

**Unified tasks.**   To seamlessly evaluate the model on new unseen tasks, it is essential to reformulate all tasks in the same way. For sequence-to-sequence frameworks, this can be done via prompting, where each task is specified by a particular textual instruction. In addition, discriminaive tasks can be cast to generation ones, and thus having only sequence generation output.

**Unified training objective.**   Due to the success of next token prediction in LLMs, it is common to use this objective to train also unified models. An alternative, is to use an equivalent to the MLM loss. The same loss is used during pretraining and finetuning.

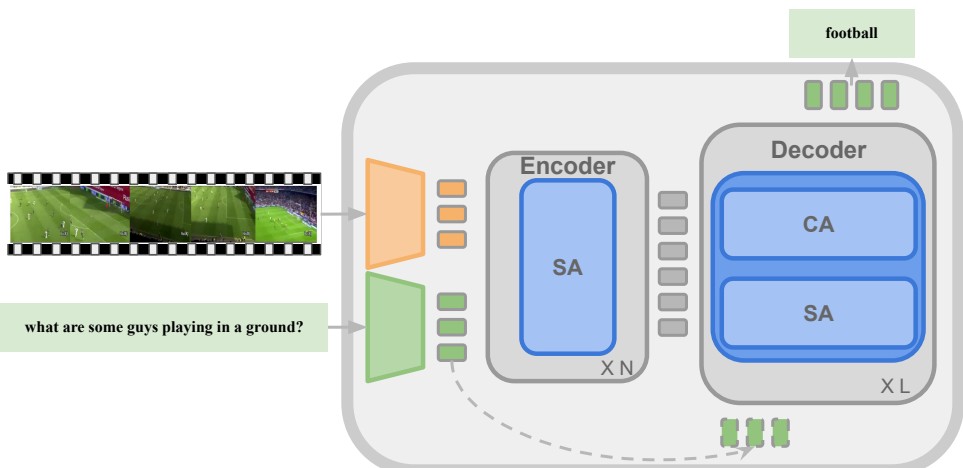

Figure 4: **UnIVAL** architecture. We use a typical encoder-decoder transformer, in addition to light-weight CNN-based modality encoders.

## D    Model Architecture

To tackle multimodal tasks at small to mid-scale, we employ an encoder-decoder LM (Vaswani et al., 2017; Lewis et al., 2020) (shown in Fig.4), as its effectiveness for multimodal tasks has been demonstrated compared to decoder-only models (Wang et al., 2021), and their superiority in zero-shot generalization after multitask training (Wang et al., 2022d). The encoder consists of stack of blocks of Self-Attention (SA), Layer Normalization (LN), GELU activations and Feed Forward Network (FFN) layers. The decoder blocks contains additionally cross-attention (CA) layers to attend to the encoder last layer tokens. Specifically, the output tokens of the encoder are considered as keys and values in the CA, while the text generated in the decoder is considered as queries. Following other approaches (Wang et al., 2022c), and to stabilize the training, we add LN layers after the SA and the FFN, and head scaling to the SA. We use independent absolute and relative position embeddings for text, images, videos and audios. We add different modality token embeddings to distinguish text from other modalities. The model parameters are initialized from BART-base model (Lewis et al., 2020).

For each modality, we use light-weight convolution architectures (*e.g.*, the encoders in orange and green in Fig.4). For images, we follow other work (Wang et al., 2021; 2022c) and use ResNet-101 trained on ImageNet. For videos, we use 3D ResNext-101 (Hara et al., 2018a) trained on Kinetics-400 (Kay et al., 2017), and for audio, we use PANN-CNN14 (Kong et al., 2020) trained on AudioSet (Gemmeke et al., 2017). We do not skip the last block in the encoders (Wang et al., 2022c), as we find that it reduces the number of tokens and accelerate the training (see Tab.18).

Each modality is encoded in the modality projection (for text we use linear embedding layer), and then concatenated to form a sequence of tokens (*e.g.*, textual and visual) before being passed to the encoder (for some tasks such as VQA, we pass also the question to the decoder). After encoding, the output of the encoder interact with the decoder via cross-attention. The decoder generates the response auto-regressively starting from a special BOS token.

## E    Pretraining Tasks

We pretrain **UnIVAL** on the following image/video-text tasks:

**Image Captioning.**    The model takes as input an image and "what does the image describe?" as text and generate a textual description of the image.

**Visual Question Answering (VQA).**   The model takes as input an image and a question and generates a textual answer based on the image.

**Visual Grounding (VGround.).**   The model takes an image and "Which region does the <text> describe?" as text and the model generates the coordinates of the bounding box described by the <text>.

**Grounded Captioning (GC).**   This is similar to image captioning, but the model should generate a description of a specific region in the image. Specifically, the model takes an image and "what does the region describe? region: <x1, y1, x2, y2>" as text and generates a caption of the region. <x1, y1, x2, y2> are coordinates of the region bounding box.

**Image-Text Matching (ITM).**   The model takes an image and a text and should predict if the text corresponds to the image. For a given image we randomly sample a caption as negative text and consider the original caption as positive. The input text is "Does the image describe <text>?" and the output is either "Yes" or "No".

**Video Captioning.**   Similarly to image captioning, the model takes a video and "what does the video describe?" and generates a video description.

**Video Question Answering (VideoQA).**   The model takes a video and question and should answer the question based on the video.

**Video-Text Matching (VTM).**   The model should predict if a text corresponds to a given video or not.

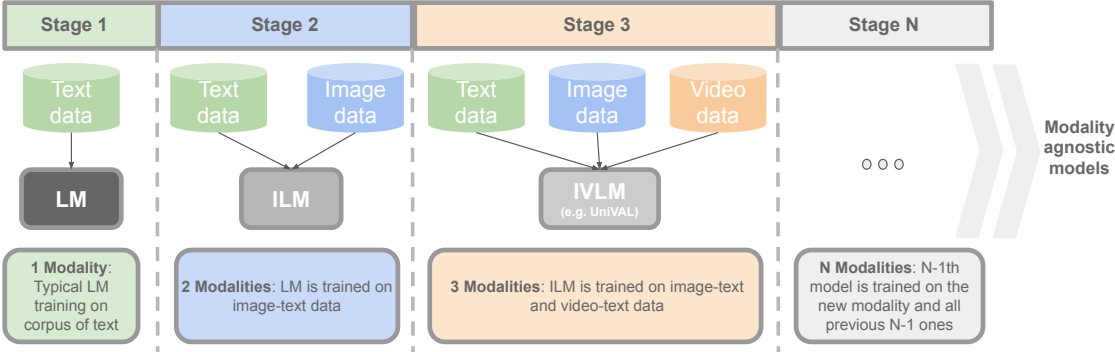

Figure 5: Multimodal Curriculum Learning. We pretrain **UnIVAL** in different stages. (1) The first pretraining is a typical training for language models on corpus of text. (2) Then, the model is trained on image and text data to obtain an Image-Language Model (ILM). (3) In the third stage, the model is trained additionally on video-text data to obtain a Video-Image-Language-Model (VILM). To obtain modality agnostic models the model should be trained on many modalities. Following this setup, **UnIVAL** can be used to solve image/video/audio-text tasks.

# F   Multimodal Curriculum Learning

Training on many tasks and modalities is computationally expensive, especially when considering long videos ore audios. To overcome this, we propose a multistage curriculum training approach (depicted in Fig.5) in which we progressively add more modalities. In stage 1, the model is trained on large corpus of text following typical next token prediction or other LM training. Thanks to the many open sourced pretrained language models, it is easier to leverage and initialize from existing LMs (*e.g.*, BART (Lewis et al., 2020) as in our case). In stage 2, the model is trained on many tasks of images and texts. Afterwards, video-text datasets are added and the model is trained on both image-text and video-text data. This is a general paradigm to efficiently train multimodal models on many modalities. Training on many tasks is more efficient, however,

the standard training on image-text alignment on image captioning can be also considered. Note that, to keep good performance on unimodal tasks, it is better to add also unimodal data.

While this training scheme is more efficient than training on all data from the beginning, using more efficient approaches from the continual learning community (Wang et al., 2023b) is extremely useful in this context, to limit the number of examples as we add more modalities, especially if the objective is to obtain modality agnostic models. Training only on the new modalities will make the model forget about previous ones.

## G Data and Implementation Details

Table 15: Downstream tasks and datasets. We show the size of different splits used in our work.

| Dataset | Modality | Task | Size (Train/Val/Test) |
|---|---|---|---|
| COCO (Lin et al., 2014) | Image-Text | Image Captioning | 113K/5K/5K |
| nocaps (Agrawal et al., 2019) | Image-Text | Image Captioning | –/4.5K/– |
| VQAv2 (Goyal et al., 2017) | Image-Text | VQA | 443K/214K/453K |
| OKVQA (Marino et al., 2019) | Image-Text | VQA | –/5K/– |
| VizWiz (Gurari et al., 2018) | Image-Text | VQA | –/4.3K/– |
| SNLI-VE (Xie et al., 2019) | Image-Text | Visual Entailment | 30K/1K/1K |
| RefCOCO (Yu et al., 2016) | Image-Text | Visual Grounding | 120K/6K/5K |
| RefCOCO+ (Yu et al., 2016) | Image-Text | Visual Grounding | 120K/6K/5K |
| RefCOCOg (Yu et al., 2016) | Image-Text | Visual Grounding | 80K/5K/10K |
| COCO (Lin et al., 2014) | Image-Text | Text to Image Generation | 80K/64K/30K |
| MSR-VTT (Xu et al., 2016) | Video-Text | Video Captioning | 6.5K/0.5K/3K |
| ActivityNet-Caption (Krishna et al., 2017a) | Video-Text | Video Captioning | 37.5K/–/17K |
| MSRVTT-QA (Xu et al., 2017) | Video-Text | VideoQA | 156K/12K/70K |
| MSVD-QA (Xu et al., 2017) | Video-Text | VideoQA | 30K/6K/12K |
| Audiocaps (Kim et al., 2019a) | Audio-Text | Audio Captioning | 47K/0.5K/1K |
| Clotho v1 (Drossos et al., 2020) | Audio-Text | Audio Captioning | 17.5K/1K/– |

### G.1 Implementation details of downstream tasks.

For image-text tasks, we keep the hyperparameters during finetuning close to those in OFA (Wang et al., 2022c). The downstream datasets are detailed in Tab.15.

**VQA.** We finetune on VQAv2 dataset and cast the task as text generation. The model is trained for 5 epochs with a batch size of 256 using Adam optimizer. We use a learning rate of $1e-4$ with linear decay and label smoothing of 0.1. The image resolution is increased to 480 and we use exponential moving average with 0.9999 decay. We use Trie based search to constraint the generated answers to the top 3.1k answers. We freeze the encoder and decoder embeddings during finetuning. The question is passed to both the encoder and decoder as prompt.

**Image Captioning.** We finetune on MSCOCO karpathy split and report standard captioning metrics. The model is trained for 4 epochs with a batch size of 128. The image resolution is set to 480 and the learning rate to $1e-5$ with linear decay. We use an encouraging (Zhao et al., 2022) cross entropy loss with label smoothing of 0.1. We freeze the encoder and decoder embeddings during finetuning.

**Visual Grounding.** We finetune on RefCOCO, RefCOCO+ and RefCOCOg for 8 epochs with batch size of 256. The images are resized to 512 and the learning rate start with $5e-5$ and decreases linearly. We train with cross entropy and label smoothing of 0.1. We limit the generation length to 4 and report the Acc@0.5.

**Visual Entailment.** The model is trained for 4 epochs with batch size of 256 and learning rate of 5e-5 that deacreases linearly. The image resolution is set to 480. The model takes only the image and the text hypothesis, without the text premise, and the generation is constrained to yes/maybe/no using Trie-based search. The text is passed to both the encoder and decoder as prompt.

**VideoQA.** The model is trained for 25 epochs on MSRVTT-QA and 40 epochs on MSVD-QA with a batch size of 128 and learning rate of $1e-4$ that decreases linearly. We sample randomly 8 frames with resolution 384. We train with cross entropy with encouraging loss and label smoothing of 0.1. We use exponential moving averaging model pass the question to both the encoder and the decoder. The answer generation is constrained to the set of possible answers via Trie-based search. We freeze the encoder and decoder embedding layers.

**Video Captioning.** We train on MSR-VTT for 15 epochs and a batch size of 256 with a starting learning rate of $1e-5$ that decreases linearly. We randomly sample 16 frames with resolution 384 and train with an encouraging cross entropy loss and label smoothing of 0.1. We freeze both the encoder and the decoder embedding layers.

**Audio Captioning.** We train for 10 epochs on Audiocaps and Clotho v1 with a batch size of 128 and starting learning rate of $1e-4$ ($5e-5$ for clotho v1). The mel bins is set to 64 and the hop size to 200. We train with encouraging cross entropy loss with label smoothing of 0.1 and freeze the encoder and decoder embedding layers.

**Text-to-Image Generation.** We follow previous work (Wang et al., 2022c) and finetune the model on the train set of MSCOCO and evaluate on 30K images from its validation set. We use the discrete tokens of VQ-GAN that are then passed to a frozen VQ-GAN decoder to generate the images. We start by training with cross-entropy loss for 50K steps and batch size of 512 ($\sim$ 60 epochs) and lr 1e-3, followed by CLIP score optimization for 5K steps and batch size of 128 and lr 1e-6. When evaluating the model we select the best image, among 24 generations based on CLIP score. We report Inception score (IS) (Salimans et al., 2016), Fréchet Inception Distance (FID) (Heusel et al., 2017) and CLIP simliarity score (CLIPSIM) (Wu et al., 2022).

# H  Parameter Efficient Fine-Tuning (PEFT): Training only the Linear Connection.

| Method | PT modality | Model size | COCO | VQA v2 val | MSR-VTT | MSRVTT-QA | Audiocaps |
|---|---|---|---|---|---|---|---|
| PromptFuse (Liang et al., 2022) | Text | 0.22B | - | 34.1 | - | - | - |
| FrozenBiLM (Yang et al., 2022) | Video-Text | 0.89B | - | - | - | 47.0 | - |
| eP-ALM (Shukor et al., 2023a) | Text | 2.8B | 97.2 | 53.3 | 50.7 | 36.7 | 63.6 |
| UnIVAL (ours) | Image-Text (S1) | 0.25B | 129.8 | 71.6 | 39.8 | 19.1 | 47.5 |
| UnIVAL (ours) | Image+Video-Text (S2) | 0.25B | 132.7 | 71.6 | 51.8 | 33.6 | 49.5 |

Table 16: **Finetuning only the linear connection on different image/video/audio-text tasks.** Despite the significantly smaller size of **UnIVAL**, the model can achieve reasonable performance when finetuned on new modalities. Scores in gray are for models pretrained on the same target modality.

Once we have powerful pretrained models, it becomes important to develop highly efficient approaches that can be adapted to various tasks and modalities. Recent studies (Shukor et al., 2023a; Merullo et al., 2022) have demonstrated the possibility of efficiently adapting unimodal pretrained models to multimodal tasks, by training *only a linear layer*. The key idea is to project modality-specific tokens onto the input text space of a language model, effectively transforming them into textual tokens, while keeping all the pretrained parameters frozen. While this approach has proven effective with large models containing billions of parameters, in this section, we explore this setup with smaller models comprising several hundred million parameters. Following **UnIVAL** pretraining, we train only the linear projection responsible for mapping the output of the modality-specific encoders to the input of the LM encoder.

As shown in Tab.16, **UnIVAL** achieves reasonable performance on new tasks and modalities despite the smaller parameter count. However, these results suggest that achieving competitive performance with only the linear connection may require larger models or training on larger datasets.

| Pretrain Modality | COCO | VQA v2 | RefCOCO+ | MSR-VTT | MSRVTT-QA |
|:---:|:---:|:---:|:---:|:---:|:---:|
| ✗ | 37.9 | 62.1 | 6.4 | 47.7 | 23.0 |
| I | 128.0 | 73.1 | 70.5 | 47.3 | 29.0 |
| V | 96.6 | 68.4 | 24.3 | 54.5 | 41.9 |
| I+V | 128.0 | 73.2 | 70.2 | 56.3 | 42.3 |

Table 17: **Knowledge transfer across modalities.** Training on images helps significantly the video tasks. However, training on videos does seem to have a significant effect on image tasks.

## I Ablation Study

**Knowledge transfer across modalities.** Here we investigate the knowledge transfer between modalities, in other words, how learning a new modality can affect the performance of the model on other modalities. We test the following hypothesis; *pretraining on more modalities should improve the overall performance on all tasks.*

Tab.17 shows that in general learning a new modality, improves the performance on other modalities. Besides, it significantly helps to solve the downstream tasks of the same modality. Compared to model initialized from scratch, training solely on image-text datasets help VideoQA. In addition, training on video-text datasets (V) significantly helps image-text tasks on VQAv2, COCO and RefCOCO+. Finally, training on both image and video-text datasets improve the performance on video-text task (w.r.t to pretraining on video) and did not degrade the performance on image-text tasks.

**Efficiency during training.** Another important aspect of our approach is the significantly shorter training time. In Tab.18, we compare the training time (finetuning for one epoch) with the previous unified model OFA (Wang et al., 2022c). Compare to OFA, our training time is significantly reduced, especially with tasks requiring high image resolution (*e.g.*, 512×512 with RefCOCO+). This is mainly due to the small number of visual tokens passed to the LM, that results from using additional convolution block in the image encoder. In addition to the training time, UnIVAL requires a $\sim 20$ GB/GPU memory when finetuned on COCO with batch size of 16. During inference (on 1 GPU AMD Instinct™ MI250X with batch size of 1), the model requires 1.2GB of GPU memory and $\sim 0.015$ seconds to generate a COCO caption.

| **Method** | COCO | VQA v2 | RefCOCO+ |
|:---|:---:|:---:|:---:|
| OFA | 5.7 | 11.5 | 1.3 |
| UnIVAL | 3.1 | 8.0 | 0.7 |

Table 18: **Estimated finetuning time in GPUh for one epoch training.** UnIVAL is significantly more efficient than OFA, especially with tasks using high image resolution. The total time is divided by the number of training GPUs.

## J Additional Results

### J.1 Text-to-image generation

| Model | Model Size | Pretrain | FID↓ | CLIPSIM↑ | IS↑ |
|:---|:---:|:---:|:---:|:---:|:---:|
| DALLE (Ramesh et al., 2021) | 12B | ✓ | 27.5 | - | 17.9 |
| CogView (Ding et al., 2021) | 4B | ✓ | 27.1 | 33.3 | 18.2 |
| GLIDE (Nichol et al., 2022) | 3.5B | ✓ | 12.2 | - | - |
| Unifying (Huang et al., 2021) | 0.2B | ✗ | 29.9 | 30.9 | - |
| NÜWA (Wu et al., 2022) | 0.9B | ✓ | 12.9 | 34.3 | 27.2 |
| OFA$^{\dagger}_{\text{Base}}$ (Wang et al., 2022c) | 0.2B | ✓ | 13.9 | 34.0 | 26.7 |
| UnIVAL (ours) | 0.2B | ✗ | 15.4 | 33.6 | 25.7 |

Table 19: Text-to-image generation on MSCOCO. Pretrain: image generation is included during pretraining.

We finetune UnIVAL on MSCOCO train set and compare the performance with other approaches. Tab.19, shows that our model is competitive with previous approaches, despite being significantly smaller in size

and does rely on image generation during pretraining. Compared to OFA, we have very close performance, especially w.r.t the CLIPSIM score.

### J.2   Linear interpolation of weights

To complement our study from the main paper, we show in Fig.6 more results when interpolating weights finetuned on different multimodal tasks. These results (on both image-text and video-text tasks) confirm those previously reported in the main paper.

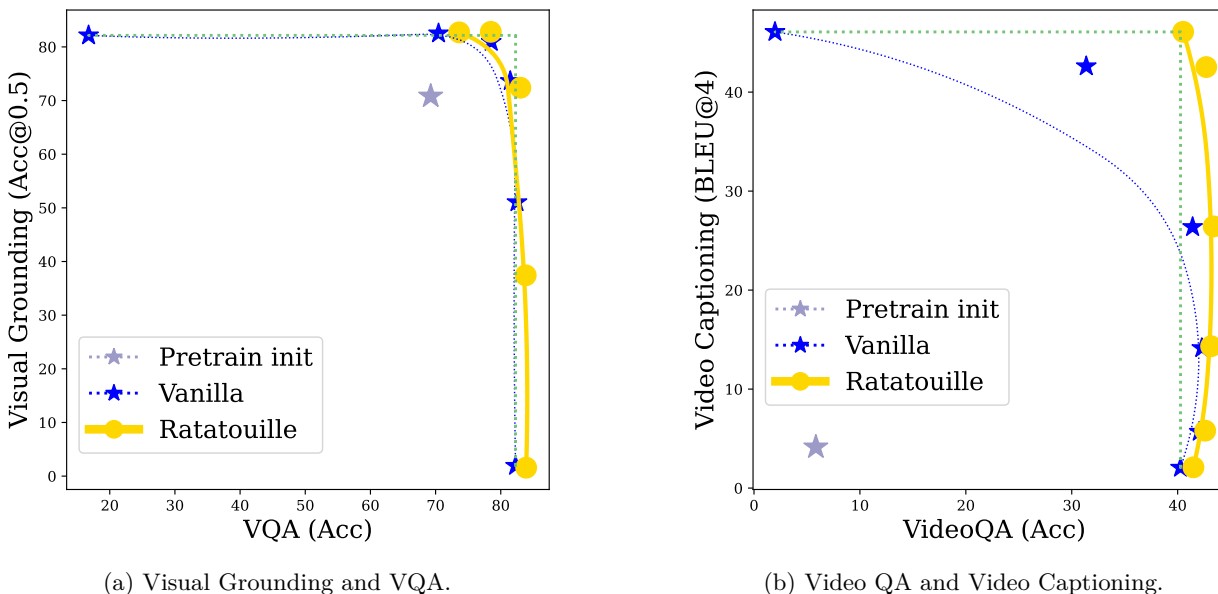

(a) Visual Grounding and VQA.

(b) Video QA and Video Captioning.

Figure 6: **Addition weight interpolation results**.

### J.3   Finetuning for OOD generalization.

Fig.20 explores the generalization abilities of different finetunings strategies after zero-shot evaluation. We evaluate the OOD performances on 3 datasets that were not seen during pretraining nor during finetuning; OKVQA (a VQA task), VizWiz (a VQA task) and nocaps (an IC task). We use the model trained on VQAv2 for OKVQA/VizWiz and on COCO Captioning for nocaps. While fusing outperforms on VizWiz the vanilla finetuning (on VQAv2), it lags behind on the other 2 evaluation datasets. Ratatouille, significantly outperforms both vanilla and fusing finetuning on all OOD datasets, which echos the observation from Rame et al. (2023a). The reason being that features diversity (promoted by delayed averaging in ratatouille) increases robustness, and thus helps OOD generalization.

| Model | OKVQA Val Acc | VizWiz Val Acc | NoCaps CIDEr (out-domain) |
|---|---|---|---|
| Vanilla | 38.06 | 13.57 | 94.39 |
| Fusing | 35.12 | 15.63 | 93.58 |
| Ratatouille | 38.97 | 18.48 | 95.28 |

Table 20: **Zero-shot evaluation of different finetunings strategy**.

## K   Discussion

In this section we discuss some of the limitations and interesting future directions.

### K.1 Limitations

**Hallucinations, abstention and other biases.** We find that our model suffers from different kind of *hallucinations* (Fig.9), however, similar to OFA, it is less inclined to hallucinate (especially after finetuning) compared to other multimodal models (Tab.21), including larger models like Flamingo (Alayrac et al., 2022) (Fig.7). Reducing hallucinations remains an ongoing challenge within the research community, which has become more prominent with the emergence of large-scale multimodal models. While certain recent studies (Biten et al., 2022; Dai et al., 2022) have proposed partial solutions to address this problem, an effective approach for mitigating hallucinations in large-scale pretrained models has yet to be established. Additionally, refraining from generating answers (Dancette et al., 2023) or visual grounding can be promising directions to enhance factuality and diminish hallucinations. Nevertheless, despite the progress made by the research community, there is still much work to be done in this area. Other biases and limitations that are crucial to address, and have not been covered in our work are; social biases, toxic generation, and explainable generation. As shown in (Shukor et al., 2023b), these limitations might not be solved by merely scaling the models and more dedicated approaches (Rame et al., 2023b; Sun et al., 2023) might be useful to address some of these issues.

| Method | CIDEr↑ | CHAIR$_S$ ↓ | CHAIR$_I$ ↓ |
|---|---|---|---|
| OSCAR$_{Base}$ (Li et al., 2020b) | 117.6 | 13.0 | 7.1 |
| VinVL$_{Larg}$ (Zhang et al., 2021) | 130.8 | 10.5 | 5.5 |
| BLIP$_{Large}$ (Li et al., 2022b) | 136.70 | 8.8 | 4.7 |
| OFA (Wang et al., 2022c) | 75.27 | **4.36** | 3.98 |
| UnIVAL | **91.04** | 4.44 | **3.64** |
| OFA Ft (Wang et al., 2022c) | **138.1** | **3.06** | **2.03** |
| UnIVAL Ft | 137.0 | 3.26 | 2.20 |

Table 21: **Hallucinations**. Comparison with different foundation models. SoTA results from (Dai et al., 2023b).

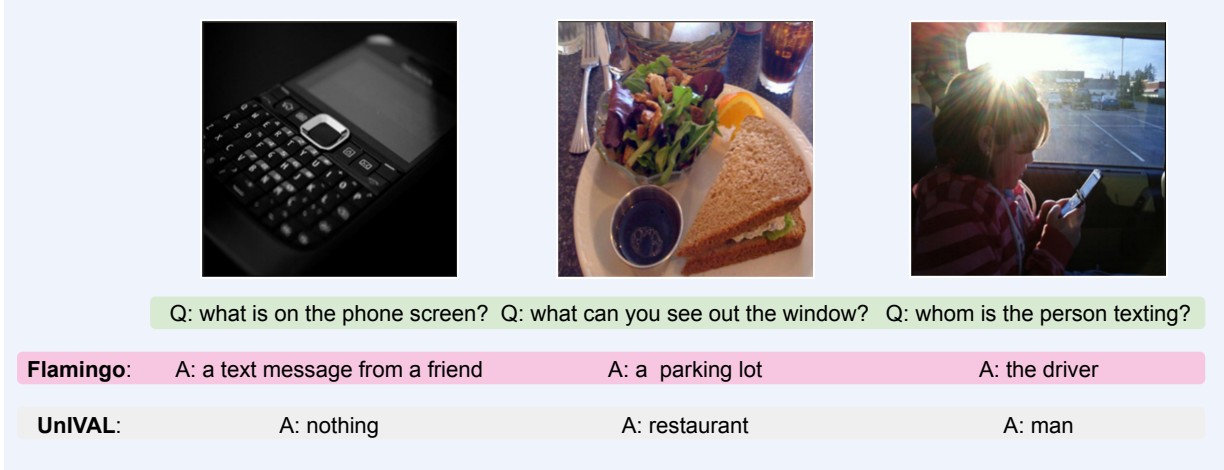

Figure 7: **Hallucinations with open-ended VQA. UnIVAL** is less prone to hallucinate compare to Flamingo-80B (Alayrac et al., 2022).

**Complex instructions following. UnIVAL** exhibits good performance when presented with straightforward instructions commonly encountered in standard benchmarks. However, it encounters difficulties when faced with complex instructions, such as delivering intricate image descriptions or providing explanations for memes. To overcome this challenge, finetuning the model using a substantial number of diverse instructions can serve as a potential solution (Xu et al., 2022; Liu et al., 2023a; Dai et al., 2023a).

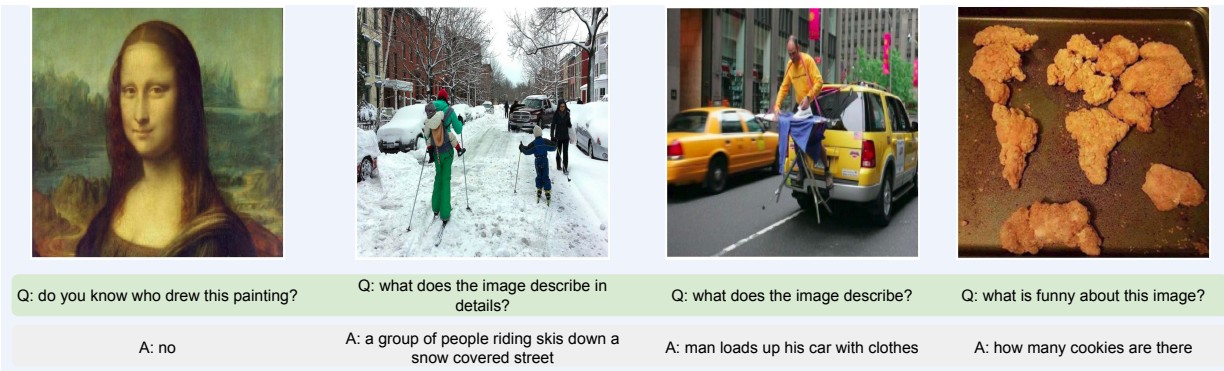

Figure 8: **Limitations of UnIVAL in following user instructions. UnIVAL** is unable to follow complex instructions.

**Unimodal tasks.** We noticed that training solely on aligned multimodal tasks can degrade the performance of the model in tackling unimodal ones. This problem is usually addressed by adding unimodal data, such as corpus of text or image, during pretraining (Singh et al., 2022; Lu et al., 2022a; Wang et al., 2022c).

**Zero-shot evaluation and efficient finetuning.** The ideal scenario is for the model to demonstrate strong performance and generalization across multiple tasks following the pretraining phase. However, we have observed that refraining from finetuning or solely training the linear connection (Shukor et al., 2023a) results in unsatisfactory performance compared to SoTA approaches. This issue can be tackled by training larger models on a greater number of instructions/tasks or by employing alternative parameter-efficient finetuning techniques (Hu et al., 2021; Lester et al., 2021).

### K.2 Future Directions

**Model scaling and better LM initialization.** In this study, we conduct experiments using a relatively small BART-initialized encoder-decoder transformer. Nonetheless, numerous intriguing language models have recently been introduced (Raffel et al., 2020; Zhang et al., 2022; Touvron et al., 2023), which could potentially enhance performance when fine-tuned for multimodal tasks. Another aspect involves reasonably scaling the model size and training it on larger datasets, which could unveil more capabilities like In-Context Learning (Dong et al., 2022) and the ability to tackle more complex tasks (Lu et al., 2022b).

**More modalities and tasks.** Our study demonstrated the feasibility of training a unified model capable of addressing tasks involving image, video, audio, and text modalities. As a result, we posit that incorporating additional modalities, either during the pretraining phase or solely during finetuning, can be accomplished straightforwardly. Furthermore, expanding the scope of tasks within each modality, such as incorporating a broader range of visual tasks (Lu et al., 2022a; Zou et al., 2023) or tasks necessitating complex reasoning abilities (Liu et al., 2023a), represents a natural extension of this work. Ideally, we hope that in the future, there will be modality-agnostic models, bridging the gap between domains and modalities.

**Towards embodied and generalist multimodal assistant agents.** Modality-agnostic models hold the potential to facilitate the development of embodied agents capable of addressing real-world challenges, including navigation and robotics manipulation, which demand the simultaneous handling of multiple modalities. Furthermore, while there has been notable progress in the NLP community regarding the construction of generalist agents, such as chatbots (Liu et al., 2023b), these advancements remain constrained in terms of their ability to accept diverse input modalities and generate outputs beyond textual form.

**Better training schemes for multitask multimodal training.** While growing the number of tasks and modalities, it is important to devise new efficient training schemes to better leverage the collaboration between

tasks, and continually support more modalities. We believe that there is more efficient approaches than our multimodal curriculum learning, to continually add more modalities while avoiding forgetting previous ones.

## L  Qualitative Results

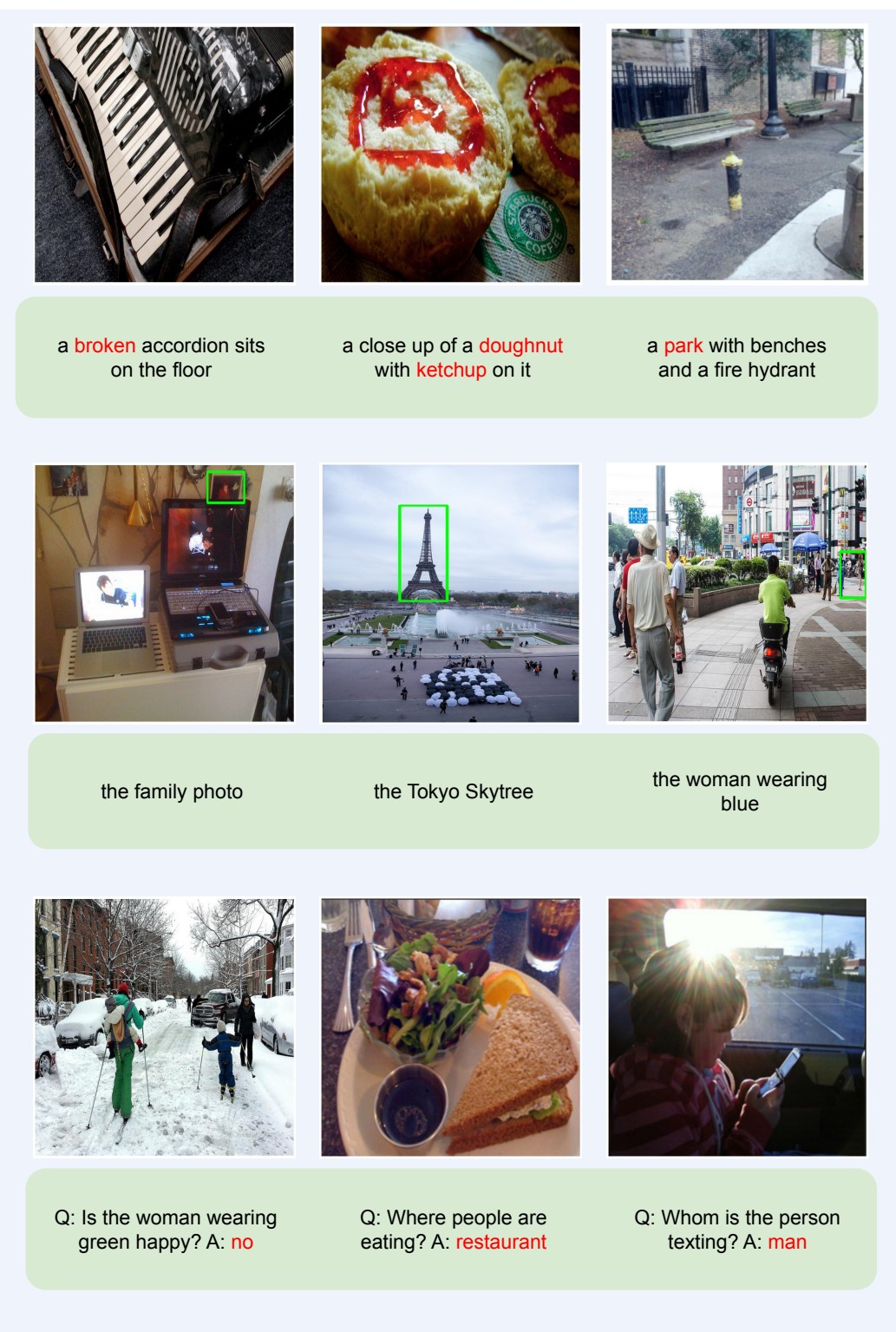

Figure 9: **Limitations of UnIVAL.** We show the limitations on different image-text tasks; (row 1) objects hallucinations (Image Captioning), (row 2) inability to capture nuanced description, object hallucinations, and struggle with far/small objects (Visual Grounding) and (row 3) answer hallucination (VQA).

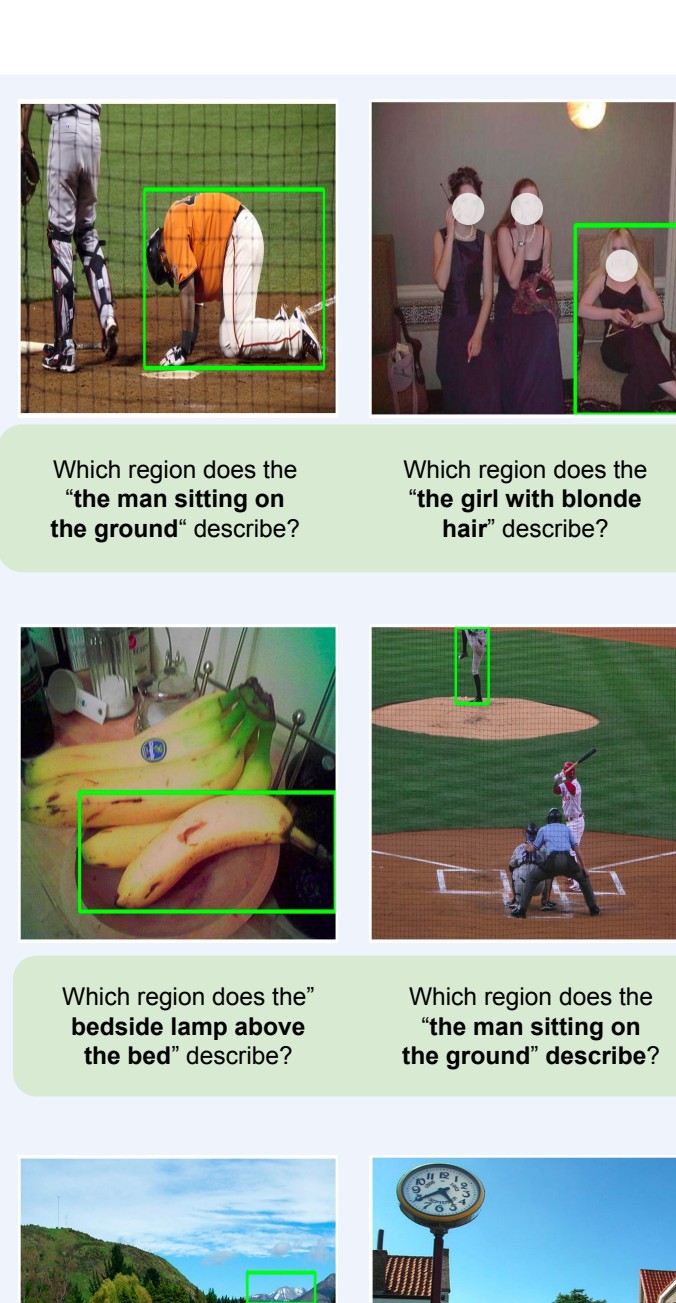

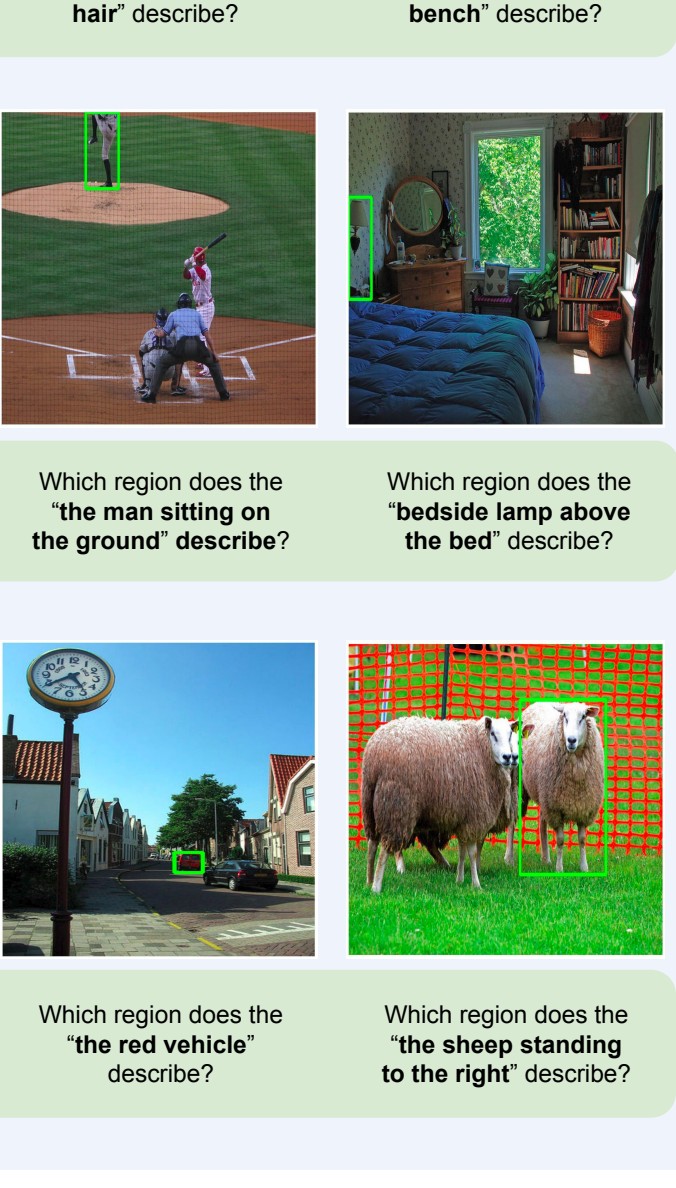

Figure 10: Visual Grounding. Image from COCO val 2014 set. Texts constructed manually.

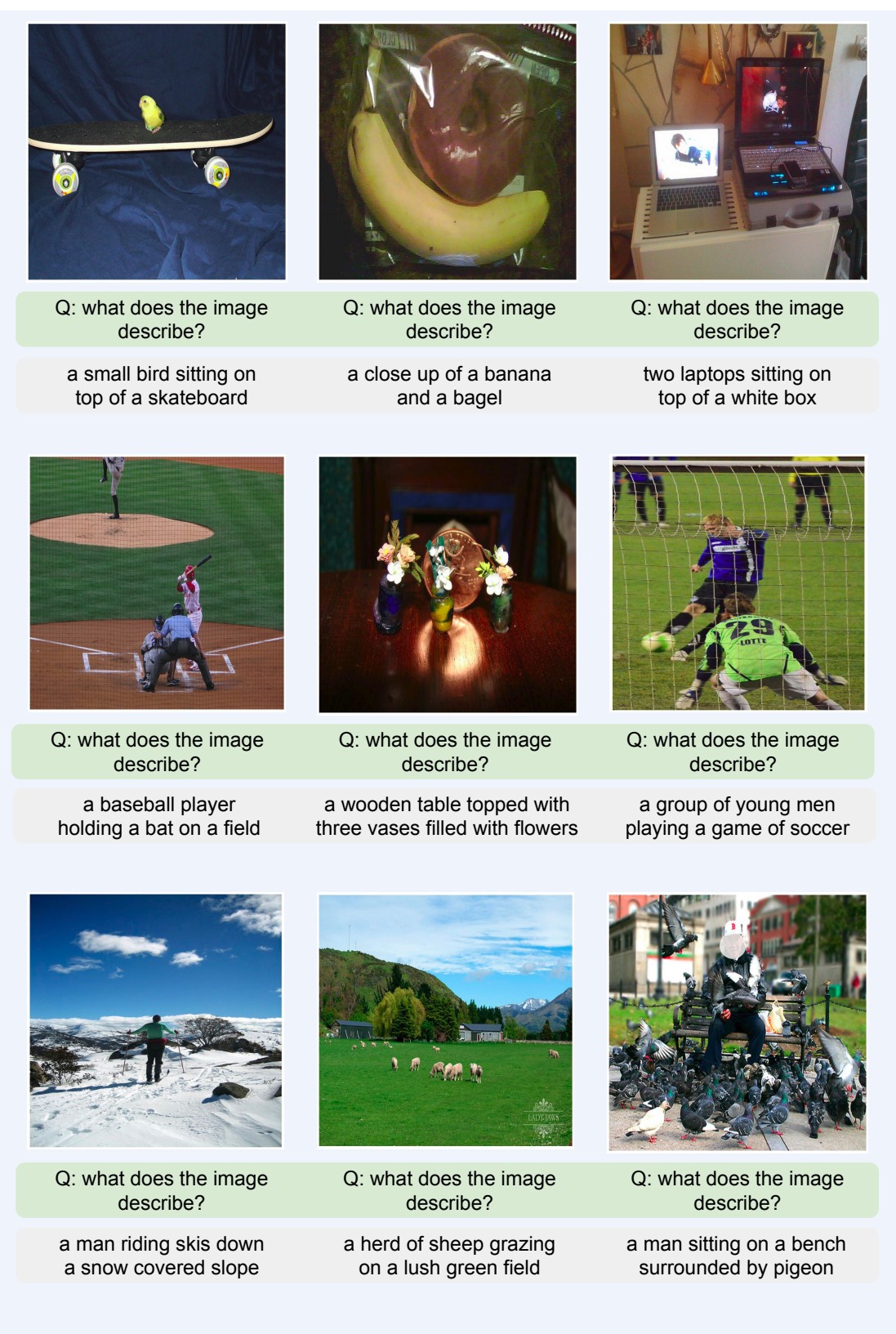

Figure 11: Image Captioning. Image from COCO val 2014 set.

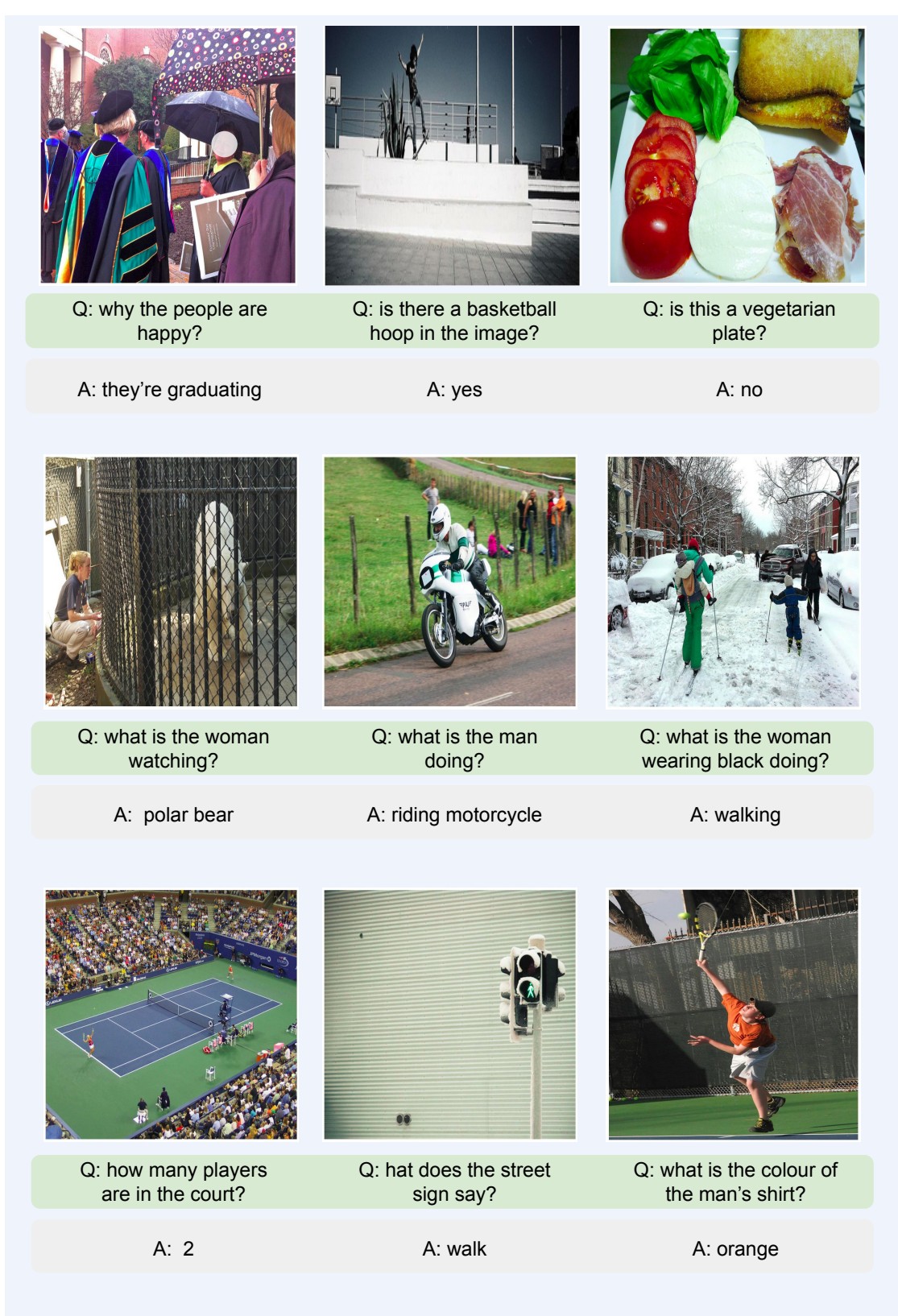

Figure 12: VQA. Image from COCO val 2014 set. Question constructed manually.

