# OpenReview forum: "UnIVAL: Unified Model for Image, Video, Audio and Language Tasks"
_TMLR — Accepted by TMLR_

### Review · Reviewer_HzJS · 2023-08-31

**Summary Of Contributions:**

The paper introduces a new unified multimodal model called UnIVAL. It is a ~0.25 billion parameter model, the core of which is pretrained on multiple vision-text and video-text tasks using small scale datasets. After pretraining, it can be used for various multimodal tasks including text, images, videos and audio such as image captioning, video captioning, audio captioning, visual question answering (VQA), video QA, visual grounding and text-to-image generation. The model achieves competitive performance to existing state-of-the-art methods across these tasks. The paper also studies merging of model weights finetuned on different multimodal tasks.

**Audience:**

Yes

**Broader Impact Concerns:**

Given that the proposed method can be extended to various tasks and datasets without requiring a lot of data, I think the authors should include a Broader Impact section to discuss these concerns. They should also point out that the model might have learnt various biases present in the datasets that were used for pretraining.

**Claims And Evidence:**

No

**Requested Changes:**

I have included all my questions that I would like the authors to answer in the weaknesses section above. The answers to all these issues are crucial to my final recommendation.

**Strengths And Weaknesses:**

**Strengths:**
- The proposed model achieves better performance than previous works for various tasks/datasets and competitive performance on other tasks/datasets.
- The proposed model should be extendable to include more tasks/datasets without many changes.
- The paper shows the importance of multimodal multitask pretraining, including task balancing and multimodal curriculum learning.


**Weaknesses:**

- My biggest concern is with the overall takeaway from the paper. The conclusions from the experiments seem a bit blurry to me after reading the paper so clarification from the authors would be very helpful. Overall, the only consistent takeaway from the experiments section seems to be that UnIVAL is better than other previous methods. But, the section fails to present a convincing argument for what exactly makes UnIVAL a better method than other works. Below are some specific cases that lead me to this high-level weakness:
    - I think the authors need to discuss explicitly and in detail the technical takeaways from this paper. Some key questions to ask — Should I just use UnIVAL over other methods for all text-vision-audio tasks? What are the cases where it is better to use UnIVAL over other methods and what are the cases where choosing other methods might be better?
    - The final goal seems to be a bit blurry to me. Is the goal to achieve a final best text-vision-audio model? If so, isn’t it better to pretrain on all available datasets? Why not also use the datasets used for zero-shot evaluation as well audio captioning and text-to-image generation (Sections 4.2 and 4.3) to obtain the best final model?
    - What are the technical differences (in terms of network architecture, loss function, etc.) from closely related works such as OFA and Unified-IO?
    - For audio captioning, Table 9, the comparison is with methods that are trained specifically for audio captioning, right? But, all of them have only seen a very small amount of data, i.e., the training dataset of the corresponding audio captioning dataset, which becomes specially relevant because these datasets are very small in size (could the authors please report the sizes?). I appreciate the novelty in applying the proposed method for a novel modality, but the proposed method is also not compared with other unified methods to compare the adaptation capability. In the current state of this section, it’s not clear what to take away from these results.
    - Table 5, Visual Grounding results, what are the results for models that are not of comparable sizes? They should also be included in the table. Also, the pretraining data is different for different models, right? Are all the compared methods pretrained on at least the data UnIVAL is pretrained on? Without this information, I am unsure what to take away from these results.
    - Why is Unified-IO not included for comparison for image-text tasks (Section 4.1.1)?
    - Why is OmniVL not included for video captioning in Table 8?


Other concerns/questions:
- The model merging across multimodal tasks seems very separate from the rest of the paper and it seems to have expected results (please correct me if the authors think otherwise). I agree that it is still useful to validate known hypotheses through experiments, but I don’t think this section is adding any contributions to the paper.
- How does merging models finetuned on different mutlimodal tasks compare to merging models pretrained on different multimodal tasks / modalities (Song et al., 2023)? I understand that it is a concurrent work but I think it should still be included, the current results seem to tell half the picture.
- “*validating the superior performances of ratatouille.*” Again, the takeaway message from this section is confusing. Why is it relevant for this paper to show that ratatouille merging is better than vanilla averaging? Same comment applies to the comparison between Fusing and Ratatouille in the first paragraph of page 12.
- For text-to-image generation task, how do you generate an image as the output?
- Why use beam search for VQA? Are the results different with all-candidate evaluation?
- Table 12: why are the two image-text multitask pretraining numbers different?
- “*Tab.13 shows that pretraining on either image-text or video-text data helps to get additional improvement*” The numbers in the table don't show that pretraining on video-text data helps.
- There are various minor issues that are enough to break the readability of the paper. Some are listed below:
    - “*training on VQA helps to get +1.9 points on Captioning and 4 points for Visual Grounding*”. Should this be 5 points for Visual Grounding if I’m reading the table correctly?
    - “*This setup is similar to the zero-shot evaluation, except that the evaluation tasks are seen during pretraining*”. This is very confusing — since the tasks are seen during pretraining, this setup is far from zero-shot. Please rephrase.
    - “*the CIDEr score of the model for λ = 0.8 with weights 0.8*Cap+0.2*VQA is 138.51 vs. 136.52*” It should be Visual Grounding instead of VQA, right?
    - “*(ii) then we launch multiple finetunings on VG from these auxiliary weights; (iii) we uniformly average all the weights finetuned on VG to obtain W1*” It should be VQA instead of VG?

---

> ### Author Response · Authors · 2023-10-14
> **Response to Reviewer HzJS**
>
> We thank the reviewer for the detailed review and appreciate the helpful comments. We try to address the expressed concerns below.
>
> # 1. Clarification of takeaways
> The main goal of the paper is not to get the best performances compared to other foundation models, but rather, to propose a **small-scale unified approach that can support/generalize to many modalities**, a set of features that are usually possible with large scale ones. Another goal, is to study the **synergy between different tasks** during pretraining (multitask training) and finetuning (multimodal model merging). In the following we will clarify more  takeaways.
>
> ## *a) Rev: What are the cases where it is better to use UnIVAL over other methods?...*
> The main takeaways is that with only  0.25B parameters and small public datasets, we can build a unified model that can support and generalize to many modalities, including image/video/audio-text.
>
> **When to use UnIVAL**: When the goal is to have **a general model that is good at many tasks, across modalities, while being limited in terms of computation resources**. Additional practical advantages is the simplicity of the model (unification across many axes) that makes it easier to scale, and adapt to new tasks and modalities. The importance of such models are more explained in the introduction (e.g., “Providing unification across…” Page 2). If the goal is to have the best performance and being unified, a larger version of UnIVAL (model/data size) might be needed.
>
> **When to use other models**: If the goal is to only target a specific task/modality and get the best performance, it might be better to focus on more specific/customized approaches.
>
> ## *b) Rev: Is the goal to achieve a final best text-vision-audio model? If so, isn’t it better to pretrain on all available datasets?*
> The realistic scenario that UnIVAL tackles is when we want to build **a model that is good at current tasks and modalities, but also works well for potential new tasks and modalities**. That is why we kept-out datasets during pretraining, to simulate such scenarios in terms of tasks (zero-shot eval on new datasets) and modalities (no pretraining on audio-text).
>
> ## *c) Rev: What are the technical differences from closely related works such as OFA and Unified-IO?*
> These models adopt unified architectures based on transformers with few differences; compared to image-text OFA, the perceptual CNN projections are different (additional layers in the image encoder to reduce the number of output tokens, and the new video and audio encoders). Compared to the image-text Unified-IO, there are additional differences, mainly the language model (BART instead of T5) and the image projection (CNNs instead of linear layer). However, the main difference is during training as our approach adopts the balanced multitask learning and the multimodal curriculum learning and pretrains on both image-text and video-text tasks. The details about the architecture are in appendix D.
>
> ## *d) Rev: the proposed method is also not compared with other unified methods to compare the adaptation capability...*
> Some of these approaches (Mei et al 2021 and Liu et al 2022 in Tab.9) leverage a pretraining on AudioSet in one way or another. We did not find other published and comparable unified models that evaluate on these benchmarks, most of current unified models such as OFA and Unified-IO are tailored for image-text tasks.
> The interesting takeaway is that without audio-text pretraining, our unified model generalizes well to audio-text tasks and competes with other customized approaches. This suggests that leveraging UnIVAL for other new modalities is possible. The dataset sizes are reported in Table 15 in the appendix.
>
> ## *e) Rev: Table 5, Visual Grounding results, what are the results for models that are not of comparable sizes?*
> It is common to compare these baselines on this benchmark (e.g. OFA paper). However, most of existing approaches are not comparable, in terms of pretraining data, model size, unified or not .... The only comparable work to ours is OFA (that pretrain on more data, including our image-text datasets), where we can notice the benefits of our multitask multimodal pretraining. Other foundation models with comparable sizes are OFA Huge (~1B param) with the following scores: 92.04 94.03 88.44 87.86 91.70 80.71 88.07 88.78 (in the same order as Tab.5).
>
> ## *f) Rev:  Why is Unified-IO not included for comparison for image-text tasks (Section 4.1.1)?*
> Unified-IO underperforms on this setup, as it is not finetuned on these benchmarks (e.g. Unified-IO scores are 61.8 (VQAv2) 104.0 (COCO)), for  fair comparison we did not include the scores. We provided a fair comparison with Unified-IO in Table 10 and 11 without doing finetuning for UnIVAL
>
> ## *g) Rev: Why is OmniVL not included for video captioning in Table 8?*
> As far as we know there are no reported results for Video Captioning on MSRVTT and ActivityNet-captions in the OmniVL paper.

---

> ### Author Response · Authors · 2023-10-14
> **Response to Reviewer HzJS (part 2)**
>
> # Other concerns/questions:
>
> ## *a) Rev: The model merging across multimodal tasks seems very separate from the rest of the paper...*
> We gently disagree with the reviewer on this point. First, multimodal weight averaging is not trivial or expected to work, as most previous approaches focus on averaging models trained on a shared task for a single modality (e.g. models trained on different classification datasets). **Weight averaging with very different tasks (VQA vs Captioning vs Visual Grounding) in a multimodal context makes such hypotheses more challenging**. Multimodal model averaging can be seen as **an alternative approach for standard finetuning** to leverage auxiliary knowledge to help generalize well. Third, as one of the goals of the work is to **explore the synergy between multimodal tasks, model merging explore this via weight interpolation**, leveraging the diversity of multimodal tasks to further improve final performance, while during pretraining we explore this with multitask training. This is only made possible thanks to our unified architecture.
>
> ## *b) Rev: How does merging models finetuned on different mutlimodal tasks compare to merging models pretrained on different multimodal tasks / modalities (Song et al., 2023)?*
> Our work is different from Sung et al. as each method tackles different problems. They focus on merging modality-specific transformers/modules (vision and text encoders), inside a multimodal model (that contains vision/language and cross-modal layers/modules), directly after pretraining (and then finetune the merged model on downstream tasks) with the  goal of reducing the model parameters. Our work focuses on the **downstream finetuninig stage, where we merge models finetuned on different multimodal tasks (VQA, Captioning, Grounding …) with the goal of leveraging the collaboration and diversity of these tasks to help generalization**. As we use a unified model (parameters shared for all modalities) the application of the work of Sung et al to our model (assuming modality-specific transformers inside the architecture) is not straightforward and then the direct comparison is not possible.
>
> ## *c) Rev: Why is it relevant for this paper to show that ratatouille merging is better than vanilla averaging?*
> One of the main benefits of our unified model is to leverage the synergy between tasks, either during multitask pretraining or model merging during finetuning. To validate the latter we tried different existing weight averaging approaches. One of the goals of this paper is to show that existing weight averaging works well in this context and almost the same observations hold for multimodalities where the WA can leverage the rich and diverse auxiliary tasks. As ratatouille surpasses other approaches in the general WA context, we wanted to show that this is still the case for this work.
>
> ## *d) Rev: For text-to-image generation task, how do you generate an image as the output?*
> We generate the images as discrete tokens and then pass these tokens to a frozen pretrained VQ-GAN decoder similar to other work (OFA, Unified-IO). Will be detailed in the revised paper.
>
> ## *e) Rev: Why use beam search for VQA? Are the results different with all-candidate evaluation?*
> All-candidate evaluation gives slightly better results as shown in the OFA paper, however ,we find it very slow to be useful in practice.
>
> ## *f) Rev: Table 12: why are the two image-text multitask pretraining numbers different?*
> The first 2 lines are trained only for 10 epochs, the other for 20 epochs
>
> ## *g) Rev: Tab.13 shows that pretraining on either image-text or video-text data helps to get additional improvement” The numbers in the table don't show that pretraining on video-text data helps.*
> Indeed, training on image-text is more helpful for text to image generation, we will clarify this more in the text.
>
> ## *h) Rev: There are various minor issues that are enough to break the readability of the paper. Some are listed below*
> Thanks for these comments. We will fix them in the revised version of the paper.
>
> ## *i) Rev: Broader Impact Concerns*
> Discussion about broader impact and limitations about biases will be added in the revised version of the paper.

---

### Review · Reviewer_hRUp · 2023-09-19

**Summary Of Contributions:**

This work proposes a new transformer model which supports multiple modalities including video audio images of text. The model is relatively compact and exhibits strong performance on a number of benchmarks. The model in code is expected to be open source. The paper describes a numbers of approaches to training which enhanced the ultimate outcome. It also explores different approaches to fine-tuning which enables model to perform well across multiple fine-tuned downsteam tasks jointly. Exploration of weights interpolation techniques such as ratatouille to enhance fine-tuning across tasks and modalities provide validation that these can be made practical beyond SOTA in a wide range of multimodal settings. The paper provides a detailed description of the data sources and of the limitations of the model in appendix.

**Audience:**

Yes

**Broader Impact Concerns:**

No specific concerns. The model card and examples provide it describe the models limitations well.

**Claims And Evidence:**

Yes

**Requested Changes:**

Please provide some systems characterization of the model in terms of speed and memory.

**Strengths And Weaknesses:**

Strength: this model is a useful contribution to the public domain. There are not many models available today that combine multiple modalities and are small enough to be easily accessible. Do a description of the training methodology and of the various approaches to fine-tuning across tasks provide interesting data points to the lore of training large multimodal models. The paper is easy to follow and well illustrated.

Weaknesses: to fully characterizes the model, it would be very useful to provide a more comprehensive systems view of the model. Statistics about entrance speed. Statistics about fine-tuning speed. High watermark memory consumption. One could argue that this is largely a systems paper assembling known techniques and mostly relying on the implementation details and careful benchmarking as the novelty.

---

> ### Author Response · Authors · 2023-10-14
> **Response to Reviewer hRUp**
>
> We would like to thank the reviewer for appreciating the work and finding our contributions useful. Below we try to address the concerns.
>
> ## *a) Rev: ... One could argue that this is largely a systems paper assembling known techniques and mostly relying on the implementation details and careful benchmarking as the novelty.*
> We gently disagree with the reviewer that our work can be viewed as a combination of known techniques. The novelty of the work includes first, proposing a **small scale unified model across many modalities**. Second, **the study of the synergy between tasks and modalities and the generalization** to novel modalities (without audio pretraining) and tasks (text-to-image generation). Third, the **multimodal model merging across multimodal tasks**, showing that we can interpolate very different tasks such VQA/Captioning and Visual Grounding to have better overall performance on in and out of distribution setups (we are the first to propose this, as far as we know).
>
> ## *b) Rev: Please provide some systems characterization of the model in terms of speed and memory.*
> As requested, in addition to Tab.2 and Tab.18 (appendix), here we provide the time/memory statistics during training and inference of UnIVAL (1 GPU AMD Instinct™ MI250X, batch size =1), we consider the finetuning on COCO:
> * GPU Memory (Training/Inference): 6.3GB (20.8GB with batch size =16)/1.2GB
> * Time (training update/generation during inference): 0.1s (0.5 batch size =16)/0.015s

---

### Review · Reviewer_wPdr · 2023-10-07

**Summary Of Contributions:**

In this paper, the authors propose UnIVAL, a transformer encoder-decoder based model presenting unified input/output and modeling across varying tasks and modalities (**I**mages, **V**ideo, **A**udio, and **L**anguage). UnIVAL casts tasks to natural language inputs (e.g., “What does the video describe?” for video captioning) and connects modality-specific encoders to a shared representation space via a process of flattening, tokenization, and projection. Unlike other work, this allows UnIVAL to provide a single task and modality-agnostic training objective: next token prediction. And, unlike many contemporary multi-modal models, UnIVAL utilizes curated task-specific datasets instead of web-scale training and requires significantly fewer parameters. With this model, the authors demonstrate that multimodal curriculum learning (MCL) with task balancing can improve pretraining efficiency by sequentially adding new modalities, instead of beginning with all available, and balancing the occurrence of different tasks within each batch. The authors demonstrate that UnIVAL with MCL has competitive performance on various image-text, video-text, and audio-text tasks, without audio ever being seen in the pretraining data. Finally, the authors show that weight interpolation can efficiently merge the skills of different fine-tuned UnIVAL models.

**Audience:**

Yes

**Broader Impact Concerns:**

I do not have major broader impact concerns for this work. The authors include a detailed model card and discussion of bias implications in the appendix. Including some of the bias discussion in section 6 (Discussion) would be appreciated but is not critical to my decision.

**Claims And Evidence:**

Yes

**Requested Changes:**

**Critical**
- Multiple statements need further justification. I don’t expect major revisions, but some addition is needed.
  - *Section 3.1 “To tackle multimodal tasks at small to mid-scale, we employ an encoder-decoder LM, **due to its effectiveness for multimodal tasks and zero-shot generalization** after multitask training.”* There is insufficient justification for this, and it is not necessarily common knowledge. Please provide some support either in citation, intuition, and/or experimentation.
  - *Section 3.1 “Another advantage of this architecture is the inclusion of bidirectional attention mechanisms in addition to unidirectional causal attention. **This is particularly beneficial for processing various non-textual modalities**.”* There is insufficient justification for this, and it is not necessarily common knowledge. Please provide some support either in citation, intuition, and/or experimentation.
  - *Section 6 “This distinction can be attributed to using **smaller LM, a component that is known to be particularly susceptible to this issue when scaled**.”* There is insufficient justification for this, and it is not necessarily common knowledge. Please provide some intuition or a citation behind this.
  - *Section 3 “We hope that the quality mitigates the need for massive datasets, thereby reducing computational requirements, **while enhancing the model’s generalization capabilities to novel tasks**.”* Similar issue to the above. This one is not critical, as it’s speculation about the effects of the author's proposed method, rather than a direct claim.
- What exactly is the “synergy” between tasks and modalities? It is referenced as something separate from knowledge transfer between tasks in the third bullet point under contributions. Either different language should be used, or the term should be further specified. For example, Table 4 shows “the synergy between different tasks and datasets.” What specifically does it mean to show “synergy” in this case?

**Non-Critical**
- What is the training procedure used for evaluating multimodal task balancing in section 3.5? If it Is the same as the standard training procedure described further on, briefly specifying this would be appreciated.
- Move some discussion of social bias impacts from the appendix into the main paper body, as it is not obvious that they are sufficiently discussed from the main paper alone.
- Typos:
  - Section 6, “attributed to using smaller LM”: should be “attributed to using **a** smaller LM”
  - Appendix, “Multimodal pretraining”: “try to learn shared and aligned latent space” should be “try to learn **a** shared and aligned latent space”

**Strengths And Weaknesses:**

**Strengths**
- The proposed method achieves competitive results with more modality-specialized methods without using web-scale data or extremely large - The authors conduct significant testing of the model on different tasks, modalities, and types of seen or unseen data. It is overall comprehensive and convincing of the method's effectiveness.
- The authors demonstrate that multimodal curriculum learning and task balancing can help improve efficiency when training multi-modal models, a useful finding for others training similar models in the future.
- The paper is well-written and easy to follow. The appendix contains significant depth that was able to answer most questions I had from the prose-heavy main paper.

**Weaknesses**
- Some statements made by the authors are lacking in justification. See the first set of requested critical changes below.

---

> ### Author Response · Authors · 2023-10-14
> **Response to Reviewer wPdr**
>
> We thank the reviewer for appreciating the work and considering our findings convincing and useful for other future work. Below we try to address some of the concerns.
>
> # Statements clarification
>
> ## *Rev: Section 3.1 “To tackle multimodal tasks at small to mid-scale, we employ an encoder-decoder LM, due to its effectiveness for multimodal tasks and zero-shot generalization after multitask training.” *
> This statement is based on two papers cited in the appendix, we move these citations to the main paper. Please check the revised version (highlighted in yellow)
>
> ## *Rev: Section 3.1 “Another advantage of this architecture is the inclusion of bidirectional attention mechanisms in addition to unidirectional causal attention. This is particularly beneficial for processing various non-textual modalities.”*
>
> In addition to the two references mentioned in the previous point. Using bidirectional attention to tackle different modalities (ViT for images, TimeSformer for videos, ASt for audios) is common. In the context of multimodal tasks, many previous works also adopt bidirectional attention (encoder-decoder) such as FLAVA, METER,  and ALBEF. In particular the recent FrozenBiLM paper (section 4.3) proposed a comparison between encoder-decoder (with bidirectional attention) and decoder-only and shows the former is significantly better on video-text tasks.
>
> ## *Rev: Section 6 “This distinction can be attributed to using smaller LM, a component that is known to be particularly susceptible to this issue when scaled.”*
>
> As there is no paper that specifically study the effect of scale on hallucinations, we rectify this statement and add supporting citations: “The reason why models such as Flamingo hallucinate more might be due to using frozen LLMs, a component that is known to be particularly susceptible to hallucinate (Zhang et al. 2023)”. Hallucinations for multimodal models based on LLMs are explored in more details in a recent paper (EvALign-ICL).
>
>
>
> ## *Rev: Section 3 “We hope that the quality mitigates the need for massive datasets, thereby reducing computational requirements, while enhancing the model’s generalization capabilities to novel tasks.”*
> This is changed to “while enhancing the overall model capability” which is  supported in Table 4 and the discussion in “Knowledge transfer across tasks and modalities” paragraph.
>
> ## *Rev: What exactly is the “synergy” between tasks and modalities?*
> By synergy we mean positive transfer, however in the paper we use transfer and synergy interchangeably. We will improve the wording in the revised version.
>
> # Other non-critical concerns:
>
> ## *Rev: What is the training procedure used for evaluating multimodal task balancing in section 3.5? If it Is the same as the standard training procedure described further on, briefly specifying this would be appreciated*
> It is trained on less epochs (10), the number of examples are balanced per task (VQA/Grounding/Captioning), one stage (no MCL), we don’t upsample image resolution during the last epoch of the training, besides that we use the same implementation details as in subsequent experiments (except the difference in pretraining data)
>
> ## *Rev: Move some discussion of social bias impacts from the appendix into the main paper body, as it is not obvious that they are sufficiently discussed from the main paper alone.*
> We will add some discussion about social biases in the revised version of the paper.
>
> ## *Rev: Typos:*
> Thanks for pointing out, we will fix them in the revised version of the paper.
>
> #### References:
>
> FLAVA: Singh, Amanpreet, et al. "Flava: A foundational language and vision alignment model." Proceedings of the IEEE/CVF Conference on Computer Vision and Pattern Recognition. 2022.
>
> ALBEF: Li, Junnan, et al. "Align before fuse: Vision and language representation learning with momentum distillation." Advances in neural information processing systems 34 (2021): 9694-9705.
>
> METER: Dou, Zi-Yi, et al. "An empirical study of training end-to-end vision-and-language transformers." Proceedings of the IEEE/CVF Conference on Computer Vision and Pattern Recognition. 2022.
>
> Frozenbilm: Yang et al. "Zero-Shot Video Question Answering via Frozen Bidirectional Language Models". NeurIPS 2022.
>
> EvALign-ICL: Shukor, Mustafa, et al. "Beyond Task Performance: Evaluating and Reducing the Flaws of Large Multimodal Models with In-Context Learning." arXiv preprint arXiv:2310.00647 (2023).

---

> > ### Comment · Reviewer_wPdr · 2023-10-18
> >
> > Thank you for the detailed response! I am generally satisfied with the updated manuscript and do not have remaining critical changes.
> >
> > *Rev: Section 3.1 “Another advantage of this architecture is the inclusion of bidirectional attention mechanisms in addition to unidirectional causal attention. This is particularly beneficial for processing various non-textual modalities.”*
> >
> > I agree with the reasoning behind this claim in your reply, but maintain that explicitly connecting it to the literature within the manuscript will be useful for TMLR readers.
> >
> > *Rev: Section 6 “This distinction can be attributed to using smaller LM, a component that is known to be particularly susceptible to this issue when scaled.”*
> >
> > It looks like a minor typo was introduced in this edit. In the sentence before the highlighted change "UnIVAL demonstrates **a** reduced hallucinations" should be "UnIVAL demonstrates reduced hallucinations."

---

### Decision · Action_Editor_avHp · 2023-11-27

**Recommendation:** Accept as is

**Comment:**

The proposed UnIVAL method is a unified architecture for text grounded image, video, and audio processing tasks that is trained with relatively modest computationally and data resources compared to many large-scale multimodal models. This is enabled by careful curriculum and multitask learning techniques ablated in the paper. Despite its relatively smaller size and lower data/compute budget, UnIVAL achieves competitive results with many modality-specific models. During the reviewing process, reviewers raised several requests for clarification that authors have responded to and made appropriate updates to the paper. There are several comparisons to other multimodal models that would have improved the work but do not invalidate the central claims. Reviewers generally were positive about the lower data requirements and model size the proposed work offers to the community. As such, the open sourcing of code and model weights promised in the manuscript will be important.

**Audience:**

After the discussion period, all reviewers agree that there is likely an audience for this work at TMLR and the AE agrees. The lower compute and data requirements of the model may be especially interesting to those operating in computationally constrained research environments.

**Claims And Evidence:**

After the discussion period, all reviewers agree that the claims are well supported in the manuscript. Clarification about the key takeaways should be made in the camera ready.